# Lidar ratios of stratospheric volcanic ash and sulfate aerosols retrieved from CALIOP measurements

Andrew T. Prata[1], Stuart A. Young[2], Steven T. Siems[1], and Michael J. Manton[1]

[1]School of Earth, Atmosphere and Environment, Monash University, Clayton, Victoria 3800, Australia.
[2]CSIRO Oceans and Atmosphere, Aspendale, Victoria 3195, Australia.

*Correspondence to:* Andrew T. Prata (andrew.prata@monash.edu)

**Abstract.** We apply a two-way transmittance constraint to nighttime CALIOP (Cloud-Aerosol Lidar with Orthogonal Polarization) observations of volcanic aerosol layers to retrieve estimates of the particulate lidar ratio ($S_p$) at 532 nm. This technique is applied to three volcanic eruption case studies that were found to have injected aerosols directly into the stratosphere. Numerous lidar observations permitted characterisation of the optical and geometric properties of the volcanic aerosol layers over a time period of 1–2 weeks. For the volcanic ash-rich layers produced by the Puyehue-Cordón Caulle eruption (June 2011) we obtain mean and median particulate lidar ratios of $69 \pm 13$ sr and 67 sr, respectively. For the sulfate-rich aerosol layers produced by Kasatochi (August 2008) and Sarychev Peak (June 2009), the means of the retrieved lidar ratios were $66 \pm 19$ sr (median 60 sr) and $63 \pm 14$ sr (median 59 sr), respectively. The 532-nm layer-integrated particulate depolarization ratios ($\delta_p$) observed for the Puyehue layers ($\delta_p = 0.33 \pm 0.03$) were much larger than those found for the volcanic aerosol layers produced by the Kasatochi ($\delta_p = 0.09 \pm 0.03$) and Sarychev ($\delta_p = 0.05 \pm 0.04$) eruptions. However, for the Sarychev layers we observe an exponential decay (*e*-folding time of 3.6 days) in $\delta_p$ with time from 0.27 to 0.03. Similar decreases in the layer-integrated attenuated color ratios with time were observed for the Sarychev case. In general, the Puyehue layers exhibited larger color ratios ($\chi' = 0.53 \pm 0.07$) than what was observed for the Kasatochi ($\chi' = 0.35 \pm 0.07$) and Sarychev ($\chi' = 0.32 \pm 0.07$) layers, indicating that the Puyehue layers were generally composed of larger particles. These observations are particularly relevant to the new stratospheric aerosol subtyping classification scheme, which has been incorporated into version 4 of the level 2 CALIPSO data products.

## 1 Introduction

Stratospheric volcanic aerosols are formed when explosive volcanic eruptions inject $SO_2$ gas and silicate ($SiO_2$) ash particles into the stratosphere. The volcanic $SO_2$ can subsequently convert to sulfate aerosols (radii from 0.1–1 $\mu$m) to form stratospheric aerosol clouds with their radiative effects persisting from weeks to years depending on the timing, location and amount of precursory $SO_2$ gas (Carn et al., 2016; Kremser et al., 2016). According to the observational record, stratospheric sulfates formed as a result of major volcanic eruptions can cause abrupt changes in global stratospheric aerosol optical depth (SAOD; Sato et al., 1993; Bourassa et al., 2012; Rieger et al., 2015). Following the eruption of Pinatubo (Philippines, 1991), this change in SAOD led to a warming of the stratosphere (Labitzke and McCormick, 1992) and cooling of the troposphere (Dutton and

Christy, 1992). Small-to-moderate eruptions also have the ability to perturb SAOD (Vernier et al., 2011) and the cumulative effect of enhanced volcanism over the previous decade may have induced a volcanic forcing large enough to temporarily slow global warming (Solomon et al., 2011; Ridley et al., 2014; Santer et al., 2014).

Volcanic ash particles, although more short-lived than sulfates, can cause localised shortwave heating (Gerstell et al., 1995), generate regional-scale temperature anomalies (Mass and Robock, 1982) and pose a serious threat to civil aviation (Prata, 2016). In a modelling study, Niemeier et al. (2009) found that the radiative heating due to stratospheric fine ash particles, released at high latitude (60°N), influenced the regional wind flow. They argued that the combination of weak local flow, a strong Coriolis force and thermal expansion of air due to volcanic ash radiative heating led to the generation of localised vortices. The study highlighted the importance of characterising the optical properties of volcanic ash, especially during the first few weeks of an eruption.

Satellite measurements allow us to determine how volcanic ash and sulfates (collectively referred to here as 'volcanic aerosols') interact with solar and terrestrial radiation. Since 2006, the CALIOP instrument aboard the Cloud-Aerosol Lidar and Infrared Pathfinder Satellite Observations (CALIPSO) satellite has been making global, vertically resolved, attenuated backscatter measurements of the Earth's atmosphere (Winker et al., 2010). CALIOP observations have been used to identify stratospheric volcanic sulfates (Carn et al., 2007; Thomason and Pitts, 2008) as well as volcanic ash in the troposphere (Prata and Prata, 2012; Winker et al., 2012; Prata et al., 2015) and stratosphere (Vernier et al., 2013; Kristiansen et al., 2015).

The lidar equation for elastic backscatter lidars, which governs the CALIOP return signal, includes both molecular and particulate components. While the molecular terms can be estimated or modelled from atmospheric data, we are left with two unknowns (particulate backscatter and extinction) and one equation. This problem is usually overcome, as in the Fernald algorithm (Fernald, 1984) by employing an extinction-to-backscatter ratio, which is now commonly referred to as the 'lidar ratio'.

Previously reported observations of the volcanic ash lidar ratio vary. Ansmann et al. (2010) and Groß et al. (2012) reported values in the range from 44–60 sr (at 532 nm), based on observations of the Eyjafjallajökull ash clouds in the free troposphere (∼2.5–4.5 km) over Germany. Wang et al. (2008) report lidar ratios from 42–65 sr for fine ash/sulfate mixed aerosol layers between 1.5 and 7 km produced by the 2001 and 2002 eruptions of Mt Etna. For sulfate-rich volcanic aerosols, the lidar ratio was determined to be 48 sr for volcanic aerosol layers at 16 km produced by the 2011 Nabro eruption (Sawamura et al., 2012). For the Sarychev volcanic aerosols, the lidar ratio was determined to be $55 \pm 4$ sr for layers measured between 10 and 15 km (O'Neill et al., 2012) and for Kasatochi a lidar ratio of $65 \pm 10$ sr was determined for a layer at 11 km (Hoffmann et al., 2010). Mattis et al. (2010) also retrieved lidar ratios for Sarychev and Kasatochi, reporting values in the range from 30–50 sr at 532 nm for layers observed between 14 and 18 km.

Since CALIOP is an elastic backscatter lidar, in most cases the lidar ratio must be chosen *a priori* in order to retrieve the extinction profile. Based on extensive ground-based sun photometer measurements taken from the Aerosol Robotic Network (AERONET; Holben et al., 1998), Omar et al. (2009) have defined six aerosol subtypes for use with CALIOP measurements in version 3 of the data products; clean continental, polluted continental, polluted dust, desert dust, clean marine and smoke. In the version 4 release there will also be a dusty marine aerosol type in the troposphere and there will be four stratospheric types.

The CALIOP scene classification algorithm (SCA; Omar et al., 2009), uses optical layer properties, surface type and layer height information to identify CALIOP feature layers as one of the predefined aerosol subtypes. By assigning each aerosol subtype with a characteristic lidar ratio, the extinction profile can be retrieved from CALIOP data (Young and Vaughan, 2009).

While the particulate lidar ratio ($S_p$) must be assigned *a priori* in the majority of cases, under certain conditions, the equations of Fernald et al. (1972) can be used to determine $S_p$ from CALIOP measurements. This occurs when the lidar ratio solution is constrained by an estimate of the two-way transmittance (Fernald et al., 1972; Young, 1995). Reliable estimates of the two-way transmittance are possible when sufficient clear air exists above and below a lofted cloud/aerosol layer. The transmittance method has previously been applied to optically thin cirrus layers (Sassen and Cho, 1992; Young, 1995), desert dust (Omar et al., 2010) and smoke plumes (Cook et al., 1972).

Stratospheric volcanic ash and sulfate layers are often observed as semi-transparent, laminar features (e.g. Winker and Osborn, 1992a; Vernier et al., 2013). Moreover, the stratosphere is generally free of meteorological clouds, desert dust, biomass burning and continental aerosols; providing the necessary clear-air conditions. The CALIOP backscatter signal-to-noise ratio (SNR), however, is significantly degraded by sunlight during the day. Thus, nighttime observations are generally required to perform a constrained retrieval on stratospheric volcanic aerosols.

Recently it has been shown that sulfate layers can be identified in CALIOP profiles using collocated measurements of $SO_2$ gas (Carboni et al., 2016). Since CALIOP is insensitive to $SO_2$, the underlying assumption is that volcanic $SO_2$ gas and $SO_4^{2-}$ aerosols are generally collocated. This is a reasonable assumption for the eruptions considered in the present study. Clarisse et al. (2013) showed that sulfate aerosols were detectable from the very onset of the Sarychev Peak eruption and that the infrared $SO_2$ and $H_2SO_4$ signatures were collocated in space and time for the first month. Similarly, Karagulian et al. (2010) demonstrated that the Kasatochi $SO_2$ cloud was collocated with sulfates for more than one month after the eruption.

This study uses the transmittance method and equations of Fernald et al. (1972) to characterise and explore the variability of the lidar ratio for stratospheric volcanic aerosol layers dominated by either ash or sulfate aerosols. Specifically, we present CALIOP-derived lidar ratios for the ash-rich layers produced by the 2011 Puyehue-Cordón Caulle (hereafter Puyehue) eruption and the sulfate-rich layers produced by the Kasatochi and Sarychev Peak (hereafter Sarychev) eruptions in 2008 and 2009, respectively. We use independent, passive infrared detection from the Atmospheric Infrared Sounder (AIRS) to identify volcanic ash in CALIOP profiles following the method presented by Prata et al. (2015). We also extend this method to sulfates using $SO_2$ as a proxy for $SO_4^{2-}$.

## 2 Satellite data

### 2.1 AIRS

The AIRS instrument is a part of the Afternoon-train (A-train; Stephens et al., 2002) and is aboard the Aqua satellite in sun-synchronous orbit at 705 km altitude. The AIRS spectrometer disperses upwelling radiation across highly sensitive detector arrays, which results in 2378 spectral samples (nominal spectral resolution of $\lambda/\Delta\lambda = 1200$). These high-spectral resolution measurements cover three infrared wavebands (3.74–4.61 μm, 6.20–8.22 μm and 8.8–15.4 μm; Aumann et al., 2003) and can

be used to detect volcanic ash (Prata et al., 2015) and SO$_2$ (Hoffmann et al., 2014). An individual AIRS granule comprises 90 × 135 pixels (1800 km × 2700 km) with a spatial resolution of 13.5 × 13.5 km$^2$ at nadir.

The data products used in the present study are the level 1B geolocated and calibrated radiances version 5.0.23. Only channels suitable for retrievals were used to calculate brightness temperatures (i.e. with L2_ignore flag set to zero; see http: 5 //disc.sci.gsfc.nasa.gov/AIRS/documentation).

## 2.2 CALIOP

The CALIPSO satellite is also a member of the A-train and carries the CALIOP instrument as its primary payload (Winker et al., 2010). Following closely behind Aqua (∼73 s), the space-borne lidar measures elastically backscattered light at 532 and 1064 nm using a three-channel receiver subsystem (Hunt et al., 2009). The ratio of the backscatter measured at these 10 wavelengths (i.e. the attenuated color ratio) can be used to infer information about particle size (Liu et al., 2009). The 532-nm signal is also split into two linear polarisation states, which enable depolarization measurements to distinguish between irregular (e.g. ash, ice, dust) and spherical (e.g. sulfates) particles.

The CALIOP level 1 version 4, 532-nm total attenuated backscatter profiles (L1-Standard-V4-00) were used to generate attenuated backscatter curtain plots. At a given wavelength, $\lambda$, the total attenuated backscatter profile, $\beta'_\lambda(r)$, is related to the 15 particulate and molecular components of backscatter by (Vaughan et al., 2009)

$$\beta'_\lambda(r) = [\beta_{m,\lambda}(r) + \beta_{p,\lambda}(r)]T^2_{m,\lambda}(0,r)T^2_{e,\lambda}(0,r)T^2_{O_3,\lambda}(0,r),\qquad(1)$$

where $r$ is the range from the lidar, $\beta_{m,\lambda}(r)$ and $\beta_{p,\lambda}(r)$ are the molecular and particulate backscatter profiles, respectively, and $T^2_{m,\lambda}(0,r)$, $T^2_{e,\lambda}(0,r)$ and $T^2_{O_3,\lambda}(0,r)$ are the molecular, effective and ozone two-way transmittance profiles, respectively. We note that the effective two-way transmittance profile, $T^2_{e,\lambda}(0,r)$, is related to the particulate two-way transmittance profile 20 via $T^2_{e,\lambda}(0,r) = T^{2\eta}_{p,\lambda}(0,r)$, where $\eta$ is defined here as the multiple scattering factor (Platt, 1973). The vertical resolutions of the level 1 backscatter profiles are altitude dependent and are broken down into five range intervals. For the altitudes ranges shown here (0–20 km), the relevant vertical resolutions are 30 m and 60 m for the altitude ranges from -0.5 to 8.2 km and 8.2 to 20.2 km, respectively.

Geometric and optical properties of layers were obtained from the level 2 aerosol layer (L2_05kmALay) product version 3. 25 (Version 4, level 2 data had not been released at the time of writing.) The vertical resolution was 60 m for all volcanic layer observations as they were within the 8.2–20.2 km altitude range interval. To ensure constrained conditions for the lidar ratio retrieval (i.e. clear air above and below a lofted layer with acceptable SNR), only stratospheric volcanic aerosol layers that had an extinction quality control flag equal to 1, a valid two-way transmittance measurement (i.e. $0 < T^2_e(r_t, r_b) < 1$) and a horizontal averaging value of 5 km were included in the analysis. We refer to 'valid' lidar ratio retrievals hereafter as having 30 satisfied these criteria. We note that the operational lidar ratio data (Final_532_Lidar_Ratio) were not used because we wanted to adjust the multiple scattering factor ($\eta$) in the lidar ratio retrieval presented in Sect. 3.2.

The level 2 optical products used in the present analysis are the effective two-way transmittance ($T^2_e(r_t, r_b)$), the integrated attenuated backscatter ($\gamma'_p$), the layer-integrated volume depolarization ratio ($\delta_v$) and layer-integrated attenuated color ratio

($\chi'$). All products are calculated relative to the base ($r_b$) and top ($r_t$) of a given aerosol layer. As in Vaughan et al. (2005), $\delta_v$ is calculated as

$$\delta_v = \sum_{k=top}^{base} \beta'_{532,\perp}(r_k) / \sum_{k=top}^{base} \beta'_{532,\|}(r_k), \tag{2}$$

where $\beta'_{532,\perp}(r)$ and $\beta'_{532,\|}(r)$ are the perpendicular and parallel components of the attenuated backscatter at 532 nm. The perpendicular and parallel components of attenuated backscatter make up the total attenuated backscatter at 532 nm such that

$$\beta'_{532}(r) = \beta'_{532,\perp}(r) + \beta'_{532,\|}(r). \tag{3}$$

The layer-integrated attenuated color ratio, $\chi'$, is calculated as

$$\chi' = \sum_{k=top}^{base} B_{1064}(r_k) / \sum_{k=top}^{base} B_{532}(r_k), \tag{4}$$

where, $B_{1064}(r)$ and $B_{532}(r)$ are the total attenuated backscatter coefficients corrected for molecular and ozone transmittance:

$$B_\lambda(r) = \frac{\beta'_\lambda(r)}{T^2_{m,\lambda}(0,r)T^2_{O_3,\lambda}(0,r)} = [\beta_{m,\lambda}(r) + \beta_{p,\lambda}(r)]T^2_{e,\lambda}(0,r). \tag{5}$$

In general, the 1064-nm backscattering component will be less than the 532-nm component for small particles and so the attenuated color ratio will also be small. Indeed, the attenuated color ratio is generally greater than 1 for cloud layers and is less than 1 for aerosols (Liu et al., 2009). The particulate integrated attenuated backscatter, $\gamma'_p$, is defined as

$$\gamma'_p = \int_{r_t}^{r_b} \beta_p(r)T^2_p(r_t,r)dr \tag{6}$$

and is approximated using the clear air trapezoid technique in the level 2 layer products (Vaughan et al., 2005). This quantity is used in the lidar ratio retrieval described in Sect. 3.2. Finally, the effective two-way transmittance, $T^2_e(r_t,r_b)$, is calculated by taking the ratio of the mean attenuated scattering ratio profiles over regions of clear air detected above and below the layer (Vaughan et al., 2009):

$$T^2_e(r_t,r_b) = \langle R'_{\text{below}}(r)\rangle / \langle R'_{\text{above}}(r)\rangle, \tag{7}$$

where the attenuated scattering ratio profile is defined as $R'(r) = \beta'_{532}(r)/\beta'_{m,532}(r)$. We note that only the top layer in a given profile was considered in the present study so that measurements of $T^2_e(r_t,r_b)$ were not degraded by signal attenuation introduced by overlying cloud/aerosol layers. For the top layer, the operational retrieval assumes a purely molecular atmosphere (i.e. $\langle R'_{\text{above}}\rangle = 1$), and so the effective two-way transmittance is calculated as $T^2_e(r_t,r_b) = \langle R'_{\text{below}}\rangle$. The clear air region is defined by the 'clear air analysis depth', which is determined via an iterative process in the CALIPSO level 2 feature detection algorithm (Vaughan et al., 2005). It should also be noted that $T^2_e(r_t,r_b)$ can only be calculated at 532 nm as the molecular scattering signal at 1064 nm is too small ($\sim$16 times weaker than at 532 nm).

The CALIOP level 2 profile products (L2_05kmAPro) were also used to obtain the normalised, ozone-corrected, total attenuated backscatter coefficient, $\beta'_N(r)$, which is required as input into the lidar ratio retrieval (discussed in Sect. 3.2). The reason for calculating $\beta'_N(r)$ from the level 2 operational products is so that a new value for $\eta$, more representative of volcanic ash/sulfates, can be used in the lidar ratio retrieval.

## 3 Methods

### 3.1 Volcanic aerosol detection in CALIOP profiles

In order to identify sulfate-rich aerosol layers in CALIOP profiles, we assume $SO_2$ is collocated with $SO_4{}^{2-}$ and adopt the $SO_2$ Index (SI) defined in Hoffmann et al. (2014). The SI is defined as the difference between brightness temperatures measured at 7.1 μm and 7.3 μm and exploits the strong absorption signature of $SO_2$ at 7.3 μm. It is defined such that positive values indicate the presence of $SO_2$ in the atmosphere;

$$\text{SI} = \text{BT}(1407.2\ \text{cm}^{-1}) - \text{BT}(1371.5\ \text{cm}^{-1}), \tag{8}$$

where $\text{BT}(\nu)$ is the brightness temperature measured at wavenumber, $\nu$. For detection of volcanic aerosols dominated by ash particles, we use the BTD algorithm defined in Prata et al. (2015). To be consistent with the terminology used in Hoffmann et al. (2014), the ash BTD algorithm is referred to hereafter as the Ash Index (AI). The AI is a 12-channel BTD algorithm designed to exploit the reverse absorption signature of volcanic ash from 10.4–11.7 μm and 8.8–9.2 μm;

$$\text{AI} = \text{BT}_1 - \text{BT}_2 + \text{BT}_3 - \text{BT}_4, \tag{9}$$

where

$$\text{BT}_1 = \frac{1}{4}[\text{BT}(856.44\ \text{cm}^{-1}) + \text{BT}(856.75\ \text{cm}^{-1}) + \text{BT}(857.06\ \text{cm}^{-1}) + \text{BT}(857.37\ \text{cm}^{-1})],$$

$$\text{BT}_2 = \frac{1}{4}[\text{BT}(964.25\ \text{cm}^{-1}) + \text{BT}(965.04\ \text{cm}^{-1}) + \text{BT}(965.44\ \text{cm}^{-1}) + \text{BT}(966.24\ \text{cm}^{-1})],$$

$$\text{BT}_3 = \frac{1}{2}[\text{BT}(1131.79\ \text{cm}^{-1}) + \text{BT}(1133.96\ \text{cm}^{-1})]$$

and

$$\text{BT}_4 = \frac{1}{2}[\text{BT}(1080.92\ \text{cm}^{-1}) + \text{BT}(1082.41\ \text{cm}^{-1})].$$

We note that Prata et al. (2015) also introduced a temperature threshold ($T_h$) to remove false detections due to variable surface emissivity over land; however, it became clear that CALIOP detections of weak ash layers were removed by this threshold

condition and so it was relaxed for the present study. As with the SI, the AI is defined such that positive values indicate the presence of volcanic ash.

Volcanic ash and sulfate aerosols are identified in CALIOP profiles based on collocated AIRS pixel values of the AI and SI, respectively. The collocation is achieved by calculating the minimum distance between a given CALIOP profile and the centre of each AIRS pixel. For the Puyehue case study, this set of collocated AIRS pixels is scanned for an AI greater than or equal to 1 K and SI below 1 K. These conditions were set to ensure that the volcanic aerosol layers analysed for the Puyehue case study were dominated by an ash signal and, importantly, did not exhibit an $SO_2$ signal. Similarly, to ensure that observations of volcanic layers for the Kasatochi and Sarychev case studies were dominated by sulfates (and not an ash), the algorithm required an SI greater than or equal to 1 K and an AI below 1 K. We also note that CALIOP profiles located south of 65°S were removed from the Puyehue analysis as conditions over Antarctica during the Southern Hemisphere winter (June/July) are conducive to polar stratospheric cloud (PSC) formation (Pitts et al., 2009).

## 3.2 The two-component lidar ratio solution for CALIOP

We develop our lidar ratio retrieval procedure following Fernald et al. (1972) and use the same notation as Young and Vaughan (2009) and Young et al. (2013). The elastic backscatter lidar equation for the normalised, ozone-corrected, total attenuated backscatter coefficient can be written as

$$\beta'_N(r) = [\beta_m(r) + \beta_p(r)] \, T_m^2(r_t, r) T_e^2(r_t, r), \tag{10}$$

where

$$T_e^2(r_t, r) = \exp\left[-2\eta S_p \int_{r_t}^{r} \beta_p(r')dr'\right] \tag{11}$$

and

$$T_m^2(r_t, r) = \exp\left[-2S_m \int_{r_t}^{r} \beta_m(r')dr'\right]. \tag{12}$$

Here $S_m$ and $S_p$ are the molecular and particulate lidar ratios, which are assumed to be constant throughout the aerosol layer. Following Fernald et al. (1972), Eq. (10) leads to the following first-order differential equation;

$$\frac{dT_e^2(r_t, r)}{dr} - 2\eta S_p \beta_m(r) T_e^2(r_t, r) = -\frac{2\eta S_p \beta'_N(r)}{T_m^2(r_t, r)}. \tag{13}$$

Solving Eq. (13) and rearranging for $S_p$ results in the solution of the two-component lidar equation;

$$S_p = \frac{1 - T_e^2(r_t, r_b) T_m^{2\eta S_p/S_m}(r_t, r_b)}{2\eta \int_{r_t}^{r_b} \beta'_N(r) T_m^{2(\eta S_p/S_m - 1)}(r_t, r)dr}. \tag{14}$$

Equation (14) is essentially Eq. (15) of Fernald et al. (1972), but using the notation of Young and Vaughan (2009) and the multiple scattering factor, $\eta$, has been included. Since Eq. (14) is transcendental, we apply an iterative solution to retrieve

$S_p$ (Fernald et al., 1972). In order to initialise Eq. (14), the solution to the single-component lidar equation could be used to calculate an initial estimate of the lidar ratio (Eq. (7) of Fernald et al. (1972)). However, for the top-most layer in the atmospheric column, CALIOP measurements can be used to make a reasonable approximation of the particulate component of the integrated attenuated backscatter $\gamma'_p$ (obtained from the level 2 data products) and an initial value of $S_p$ can then be obtained using

$$S_p = \frac{1 - T_e^2(r_t, r_b)}{2\eta\gamma'_p}. \tag{15}$$

This value is then substituted into Eq. (14) to calculate a refined estimate of $S_p$. The refined estimate is then compared with the previous value of $S_p$ and the iteration continues until consecutive solutions converge to within a threshold of 0.01% (Fernald et al., 1972).

### 3.2.1 Using the level 2 products to retrieve $S_p$

In order to evaluate Eq. (14), the normalised, ozone-corrected total attenuated backscatter coefficient, $\beta'_N(r)$, must be known. In order to obtain $\beta'_N(r)$ from the level 2 products, we evaluate Eqs. (10)–(12) using the operational values of $S_m$, $S_p$, $\beta_m(r)$, $\beta_p(r)$ and $\eta$. The values of $S_p$ and $\eta$ are obtained from the level 2 aerosol layer product (L2_05kmALay) and $\beta_p(r)$ is obtained from aerosol profile product (L2_05kmAPro). The molecular backscatter profile, $\beta_m(r)$, is calculated from the Global Modelling and Assimilation Office (GMAO; Rienecker et al., 2008) meteorological data provided with the level 2 aerosol profile product and $S_m$ is assumed to be a constant. Note that the molecular lidar ratio is often assumed to be $8\pi/3$. However, this does not include the effects of molecular polarisability. Additionally, the narrow bandwidth of CALIOP's optical filter means that it does not see all of the scattered wavelengths near the central elastic wavelength and the appropriate value of $S_m$ for use with CALIOP data at 532 nm is 8.70447 sr rather than $8\pi/3 \approx 8.37758$ sr.

### 3.2.2 Multiple scattering considerations

The reason for calculating $\beta'_N(r)$ from the level 2 operational products (as above) is so that $S_p$ can be re-calculated, via Eqs. (14) and (15), using a new value for $\eta$ that is more representative of volcanic ash or sulfates. The multiple scattering factor, by definition, varies from 0 to 1 (Platt, 1973). Single scattering is represented by $\eta = 1$ while lower values of $\eta$ represent increased multiple scattering. In the CALIPSO level 2 version 3 datasets, $\eta$ is set to 0.6 for all stratospheric features. However, we argue that this approximation may overestimate the effect of multiple scattering in the volcanic aerosols layers considered here. Winker (2003) demonstrated that the value of $\eta$ for aerosols was a strong function of geometric thickness. Essentially, as the geometric thickness of the aerosol layer is increased the value of $\eta$ asymptotes towards unity (layers thicker than 500 m correspond to $\eta \geq 0.85$). Given that the mean geometric thickness of the Puyehue layers was $1.82 \pm 0.55$ km (Table 1), $\eta$ was assumed to be $0.90 \pm 0.05$. Accordingly, this value was set higher than the multiple scattering factor used for the Eyjafjallajökull ash layers ($0.85 \pm 0.05$; Winker et al., 2012), which were reported to have a mean geometric thicknesses of 0.75 km (Winker et al., 2012). The multiple scattering effects of volcanic sulfates are expected to be similar to that of spherical, fine mode, sulfurous aerosols; analogous to the polluted continental aerosol subtype defined in Omar et al. (2009). For the

**Table 1.** Mean and standard deviation of the geometric layer properties for the Kasatochi, Sarychev and Puyehue case studies.

| Eruption | Number of layers | Layer-top (km) | Layer-base (km) | Layer-thickness (km) |
|---|---|---|---|---|
| Kasatochi | 140 | 13.69 ± 2.03 | 12.62 ± 2.04 | 1.06 ± 0.47 |
| Sarychev | 183 | 13.80 ± 1.85 | 12.40 ± 1.76 | 1.40 ± 0.41 |
| Puyehue | 374 | 12.45 ± 0.81 | 10.63 ± 0.63 | 1.82 ± 0.55 |

polluted continental class, multiple scattering is also expected to have a small effect on optical depth (Young et al., 2008) and, therefore, the retrieved lidar ratio. Considering also that the mean thicknesses of the Kasatochi and Sarychev layers were 1.06 ± 0.47 km and 1.40 ± 0.41 km, respectively (Table 1), $\eta$ was set to 0.95 ± 0.05 for sulfate aerosols. We also compared the re-calculated lidar ratio against the operational lidar ratio using the operational value for $\eta$ as a check on our method and found that the average difference was ∼1%.

### 3.3 Retrieving the particulate depolarization ratio

As we can use the value of $S_p$ obtained from Eq. (14) to retrieve the profile of particulate backscatter, $\beta_p(r)$, we are also able to retrieve the layer-integrated particulate depolarization ratio, $\delta_p$, which is an intrinsic property of the aerosol layer. The value of $\delta_p$ can be derived from the layer-integrated volume depolarization ratio, $\delta_v$, by adapting the approach of Tesche et al. (2009) to integrated quantities:

$$\delta_p = \frac{\gamma_m(\delta_v - \delta_m) + \gamma_p \delta_v(1 + \delta_m)}{\gamma_m(\delta_m - \delta_v) + \gamma_p(1 + \delta_m)}, \tag{16}$$

where

$$\gamma_m = \int_{r_t}^{r_b} \beta_m(r)dr \tag{17}$$

and

$$\gamma_p = \int_{r_t}^{r_b} \beta_p(r)dr. \tag{18}$$

Here the particulate backscatter profile, $\beta_p(r)$, is calculated using the retrieved 532-nm particulate lidar ratio and the numerical integration procedure of Fernald (1984). We also define $\delta_m$ as the layer-integrated molecular depolarization ratio. Due to CALIOP's narrow band optical filter, $\delta_m$ is the depolarization ratio at the central Cabannes line, which can be assumed to be a constant; $\delta_m \approx 0.003656$ (Hostetler et al., 2006).

We also note that the layer-effective particulate color ratio, $\chi_p$, can be retrieved using the two-color method of Vaughan (2004). This approach seeks to minimise a non-linear function by simultaneously varying $S_{p,1064}$ and $\chi_p$ using the method of non-linear least squares. However, for the case studies considered here, we found that the method was rather insensitive to variations in the 1064-nm particulate lidar ratio, often resulting in non-physical solutions for $S_{p,1064}$. We expect that this was

**Table 2.** Mean, median and standard deviation of the optical layer properties for the Kasatochi, Sarychev and Puyehue case studies. The symbols used for the particulate lidar ratio, particulate depolarization ratio, volume depolarization ratio and attenuated color ratio are $S_p$, $\delta_p$, $\delta_v$ and $\chi'$, respectively.

| Eruption | Number of layers | $S_p$ (sr) | | | $\delta_p$ ($\delta_v$) | | | $\chi'$ | | |
|---|---|---|---|---|---|---|---|---|---|---|
| | | Mean | Median | Std. Dev. | Mean | Median | Std. Dev. | Mean | Median | Std. Dev. |
| Kasatochi | 140 | 65.78 | 59.81 | 18.79 | 0.09 (0.08) | 0.08 (0.08) | 0.03 (0.03) | 0.35 | 0.34 | 0.07 |
| Sarychev | 183 | 63.01 | 58.96 | 13.59 | 0.05 (0.05) | 0.04 (0.04) | 0.04 (0.03) | 0.32 | 0.31 | 0.07 |
| Puyehue | 374 | 68.91 | 66.87 | 12.65 | 0.33 (0.28) | 0.33 (0.28) | 0.03 (0.03) | 0.53 | 0.54 | 0.08 |

due to the relatively weak signals and low optical depths of the volcanic aerosol layers under examination. As these results were inconclusive, and require a more complete treatment of the sources of error, we decided this analysis was outside of the scope of the present analysis and therefore do not report the results here.

## 4 Case studies and results

### 4.1 Kasatochi

Activity at the Aleutian Island volcano, Kasatochi (52.18°N, 175.51°W) began over a period from 7–8 August 2008 (Waythomas et al., 2010) with $SO_2$ detectable in the atmosphere for at least a month (Krotkov et al., 2010). Using the SI, it was found that the Kasatochi signature was detectable in AIRS measurements until 28 August 2008. All of the available nighttime CALIOP and AIRS data from 8–28 August covering a geographic region from 30°N to 90°N to 180°W to 180°E were included in the present analysis. As seen in Fig. 1a, the $SO_2$ dispersion was extremely complex, with the $SO_2$ cloud being dispersed into the atmosphere over a period of $\sim$3 weeks until it became well-mixed and undetectable by AIRS. In total, 140 valid lidar ratio retrievals were made for the Kasatochi layers. The mean layer-top height and thickness of the Kasatochi layers were $13.69 \pm 2.03$ km and $1.06 \pm 0.47$ km, respectively. The mean particulate depolarization and attenuated color ratios were $0.09 \pm 0.03$ and $0.35 \pm 0.07$, respectively, indicating observations of aerosols layers optically dominated by sulfates; composed of small, spherical particles. The mean and standard deviation of the lidar ratios for the Kasatochi layers retrieved over a time period from 8–28 August were $66 \pm 19$ sr (median of 60 sr). The lidar ratios ($S_p$) and color ratios ($\chi'$) were quite variable with time; making it difficult to infer any clear trends in these parameters. The particulate depolarization ratios ($\delta_p$) remained largely unchanged during the measurement time period (Fig. 8d). Figure 2 shows the respective distributions of the optical properties for each eruption case study. The layer-mean properties are given in Tables 1 and 2.

### 4.2 Sarychev

Sarychev (48.09°N, 153.20°E), which is one of the most active volcanoes in the Kuril Island chain (Russia), began to erupt on 11 June 2009 (Rybin et al., 2011). AIRS detected an ash and $SO_2$ signature on June 12; however, CALIOP data was not

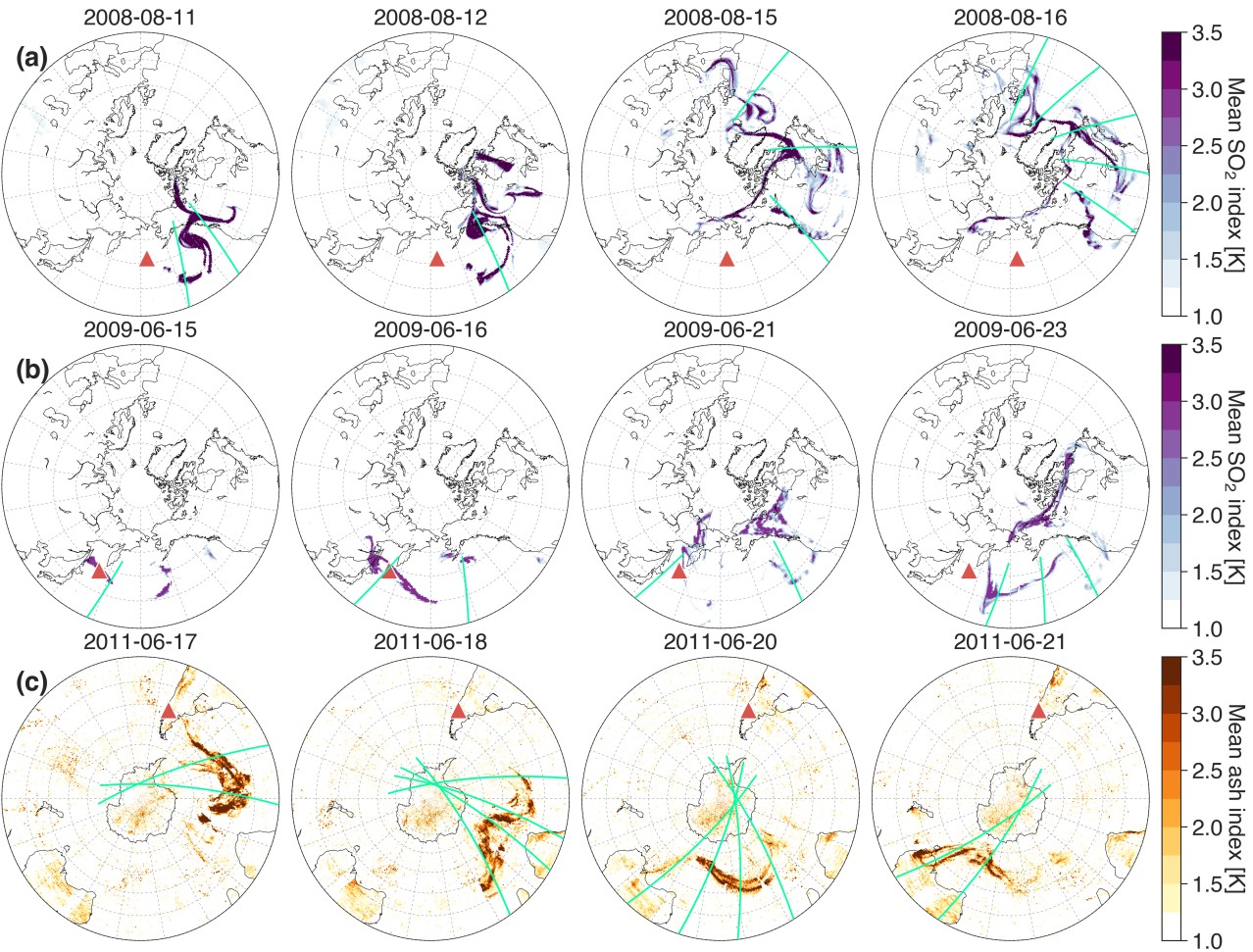

**Figure 1.** CALIOP/AIRS overview for a selected number of days for each of the case studies analysed; Kasatochi (a), Sarychev (b) and Puyehue (c). The locations of each volcano are plotted as red triangles. The AI (Ash Index) and SI (SO$_2$ Index) have been re-gridded into $0.5° \times 0.5°$ grid boxes and have been averaged by the number of data points falling into a given grid box and therefore represent AI and SI means. Over-plotted green lines indicate CALIOP overpasses that contained valid lidar ratio retrievals.

available from 12–14 June 2009. According to surface observations, no more ash or $SO_2$ was seen emanating from the volcano after 24 June, but $SO_2$ was still detectable in the atmosphere (Williams and Thomas, 2011). Data for the Sarychev case study were therefore collected from 15 June to 12 July 2009, covering the same geographic region as the Kasatochi case study. Figure 1b provides an overview of the Sarychev $SO_2$ dispersion. Unlike Kasatochi, the Sarychev $SO_2$ signature initially separated into two distinct $SO_2$ clouds that dispersed toward the east and northwest. The eastward-traveling $SO_2$ cloud remained over the Alaskan peninsula for several days, while the northwestward $SO_2$ cloud travelled south as it crossed back over the volcano. In total, 183 valid lidar ratio retrievals were obtained. The mean optical properties of the Sarychev layers shared many similarities with the Kasatochi layers (Fig. 2); however, the Sarychev particulate depolarization ratio exhibited an exponential decrease with time over 3.6 days. A similar decreasing trend was also observed for the attenuated color ratio. The time evolution of all optical properties are discussed in Sect. 6.2 and are shown in Fig. 8. The mean particulate depolarization ratio was $0.05 \pm 0.04$ and mean attenuated color ratio was $0.32 \pm 0.07$ (Table 2). The mean lidar ratio for the Sarychev layers was $63 \pm 14$ sr (median of 59 sr), corresponding to a layer-mean height and thickness of $13.80 \pm 1.85$ km and $1.40 \pm 0.41$ km, respectively (Table 1).

### 4.3   Puyehue

The eruptions of Chilean volcano, Puyehue (40.59°S, 72.12°W) began on 4 June 2011 and resulted in wide-spread and far-reaching ash layers that caused flight cancellations in Australia and New Zealand. Vernier et al. (2013) analysed CALIOP observations of the volcanic aerosols produced by Puyehue and found that the layers were primarily made up of ash particles with sulfates contributing to less than 10% of the total attenuated backscatter. In the present analysis, we avoid ice-rich layers and identify ash-rich layers using passive infrared detection from collocated AIRS pixels (i.e. AI $\geq$ 1 K and SI < 1 K). The CALIPSO analysis presented by Vernier et al. (2013) also showed that the ash clouds remained near the tropopause as they were driven around the Southern Hemisphere by a strong westerly polar jet. This spatial description of the Puyehue aerosols has been corroborated by several other authors (Klüser et al., 2013; Hoffmann et al., 2014; Theys et al., 2014).

CALIOP was switched into safe mode on 4 June, and again from 6–14 June 2011 (with 46.8% coverage on 15 June). During this time period the volcanic aerosols made their first circuit around the Southern Hemisphere. The observations included in the present analysis are therefore representative of aged ($\sim$2 weeks) ash-rich volcanic aerosol layers. The AIRS observations were analysed over a time period from 16 June to 4 July and a geographical area from 20°S to 90°S and 180°E to 180°W (Fig. 1c). The CALIOP profiles were restricted to latitudes north of or equal to 65°S to avoid PSCs (as noted in Sect. 3). In total, 374 valid lidar ratio retrievals were applied to CALIOP profiles containing stratospheric aerosol layers. The mean layer-top height and thickness of the Puyehue layers were $12.45 \pm 0.81$ km and $1.82 \pm 0.55$ km, respectively (Table 1). In contrast to the optical properties of the Kasatochi and Sarychev layers, the Puyehue layers exhibited consistently high depolarization ratios ($\delta_p = 0.33 \pm 0.03$; Table 2), indicating aerosol layers optically dominated by non-spherical particles over the measurement period. The layer-integrated attenuated color ratios for the Puyehue case study were also higher ($\chi' = 0.53 \pm 0.08$; Table 2) than the Kasatochi and Sarychev case studies ($\chi' = 0.32$–$0.35$). In general, changes in the Puyehue lidar ratios ($S_p$ mean of 69 $\pm$ 13 sr and median of 67 sr) with time were quite similar to the changes in lidar ratio with time for Kasatochi and Sarcyhev

case studies. The lidar ratio distributions for the three case studies were similar in shape and were all positively skewed. We therefore provide both the mean and median lidar ratios (annotated on each histogram of Fig. 2).

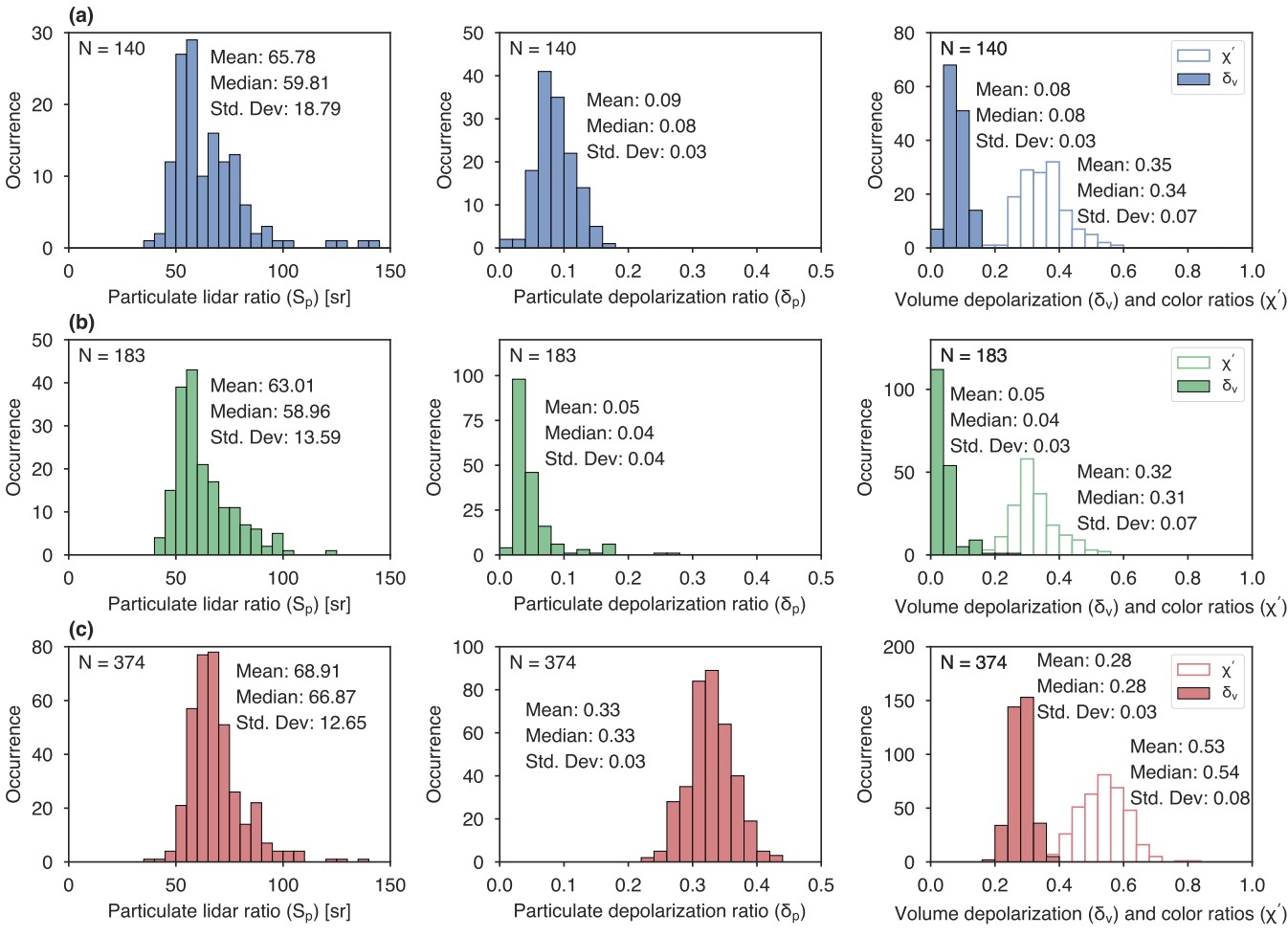

**Figure 2.** Histograms of the particulate lidar ratio (left column), layer-integrated particulate depolarization ratio (middle column) and layer-integrated volume depolarization and attenuated color ratios (right column) for the three case studies; (a) Kasatochi plotted in blue, (b) Sarychev plotted in green and (c) Puyehue in red.

## 5   Error sensitivity and propagation analysis

As discussed in Young et al. (2013), errors in a constrained retrieval of $S_p$ can be broken down into two main categories: calibration/renormalisation error, $\epsilon(\beta'_N)$, and error in the transmittance constraint, $\epsilon(T_e^2)$. We also consider possible errors in the choice of the multiple scattering factor, $\epsilon(\eta)$. We do not, however, consider the impact of random noise on the lidar

ratio retrieval. Essentially, we assume that error due to random noise will be negligibly small after 5 km averaging and thus insignificant in comparison to the other sources of error.

## 5.1 Errors in calibration/normalisation

Rogers et al. (2011) provide a comprehensive assessment of the version 3.01 CALIOP 532-nm total attenuated backscatter calibration. For nighttime measurements under clear-air conditions the mean relative error was reported to be 2.7% ± 2.1% when compared against airborne HSRL measurements. One of the main sources of error that is particularly relevant here, can arise in the case of an undetected (background) stratospheric aerosol layer. Vernier et al. (2009) highlighted how this issue would impact the CALIOP calibration region, concluding that undetected aerosols up to 35 km led to an underestimation of the particulate (aerosol) scattering ratio (an average relative error of 6%), with the effects most pronounced in the tropics (20°N–20°S). Although the observations presented here are confined to middle–high latitude regions, they directly coincide with ongoing volcanic eruption events, and so we must consider errors introduced by aerosol contamination (which have not been corrected for in the version 3 datasets).

Considering the ∼5% calibration error suggested by Rogers et al. (2011) and the 6% aerosol contamination error suggested by Vernier et al. (2009), we anticipated a relative error of 10% in the normalised, attenuated backscatter profile (i.e. $\epsilon(\beta'_N)/\beta'_N$ = 10%).

## 5.2 Errors in transmittance

The CALIOP level 2 aerosol products provide an estimate of the measured two-way transmittance error, which is calculated as the standard deviation of the attenuated scattering ratio in the clear air region below the detected layer (Vaughan et al., 2005). For the case studies considered, the means (and standard deviations) of the two-way transmittance relative errors were 16.04% ± 2.94%, 16.69% ± 2.72%, and 16.70% ± 3.84% for Kasatochi, Sarychev and Puyehue, respectively. However, since the operational algorithm (Vaughan et al., 2009) assumes pure Rayleigh scattering above the top layer of a given CALIOP profile, it is assumed that there is no attenuation by undetected layers aloft and that all of the attenuation is in the detected layer. In this case the estimate of $T_e^2$ will be too low and $S_p$ will be too high. Rogers et al. (2011) considered the possible influence of volcanic aerosols affecting the two-way transmittance between 8–30 km. Based on volcanic stratospheric optical depths from Mattis et al. (2010), they estimated a maximum bias in the two-way transmittance of 3%. Considering the mean transmittance errors for the three case studies (∼17%) and the error introduced by undetected volcanic aerosols (∼3%), a relative error of 20% in the effective two-way transmittance constraint was assumed (i.e. $\epsilon(T_e^2)/T_e^2$ = 20%).

## 5.3 Error propagation analysis

To estimate how the errors in $\beta'_N$, $T_e^2$ and $\eta$ propagate into errors in $S_p$ a multi-variable functional approach (Hughes and Hase, 2010) was applied to Eq. (14) to calculate a perturbation error for each variable. As discussed in the previous sections, $\beta'_N$ and $T_e^2$ were perturbed by 10% and 20%, respectively, and $\eta$ was perturbed by 0.05. If any variable was perturbed outside of its

physical bounds then it was set to the relevant upper or lower bound. Each perturbation error was then summed in quadrature to calculate the absolute error in the particulate lidar ratio:

$$\epsilon(S_p) = \pm\sqrt{\epsilon(S_{p,\beta'_N})^2 + \epsilon(S_{p,T_e^2})^2 + \epsilon(S_{p,\eta})^2}, \tag{19}$$

where $\epsilon(S_{p,\beta'_N})$, $\epsilon(S_{p,T_e^2})$ and $\epsilon(S_{p,\eta})$ represent the three components of error in $S_p$. The subscripts represent the variable that

5   was perturbed while holding the other two variables constant. Figure 3 illustrates, for each case study, how each of the three perturbation errors propagated into the error in $S_p$. The assumed relative errors in $\beta'_N$ and $T_e^2$, translated into mean absolute component errors of $\sim$6 sr and $\sim$14 sr, respectively, while the assumed error perturbations of 0.05 in $\eta$ corresponded to errors in $S_p$ of $\sim$3 sr. Overall, the perturbation errors, when summed in quadrature, corresponded to a mean absolute error in $S_p$ of $\sim$15 sr.

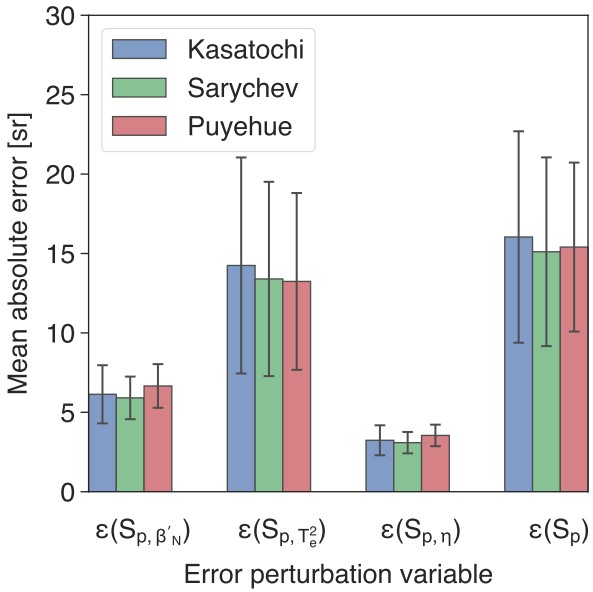

**Figure 3.** Perturbation errors for each case study; Kasatochi (blue), Sarychev (green) and Puyehue (red). The standard deviations for each perturbation error are plotted as whiskers over each bar plot.

10   As $T_e^2$ was considered to be the largest source of error in $S_p$, we examined how the relative error in the lidar ratio, $\epsilon(S_p)/S_p$, varied as a function of $T_e^2$ (Fig. 4). Here we see that the relative error in $S_p$ asymptotes toward $\sim$10% as $T_e^2$ approaches zero and increases exponentially as $T_e^2$ approaches unity. In other words, for non-transmissive (optically thick) layers, error in the retrieved value of $S_p$ will be limited by errors in $\beta'_N$ and $\eta$. For highly transmissive (optically thin) layers, error in $T_e^2$ will become the dominant source of error in $S_p$.

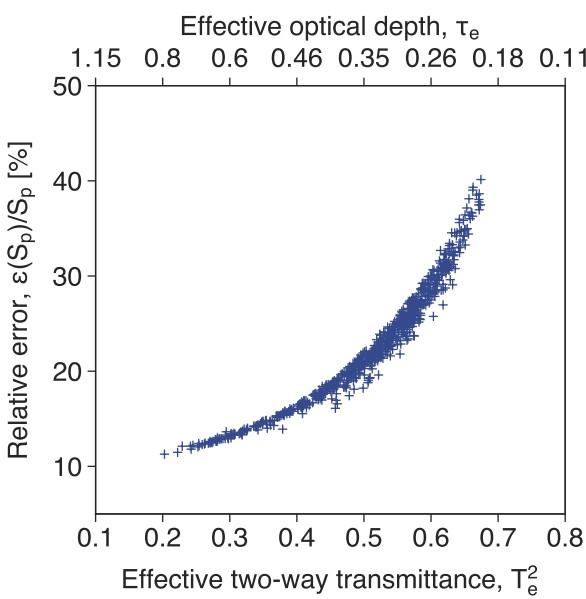

**Figure 4.** Relationship between $T_e^2$ and the relative error in the particulate lidar ratio, $\epsilon(S_p)/S_p$.

## 6 Discussion

### 6.1 Lidar ratio retrievals for selected observations

Figures 5–7 show how the CALIOP/AIRS analysis performed for an individual AIRS granule selected from each case study; illustrating the conditions under which the lidar ratio retrievals are successful and how the volcanic layers correlate with the

AI and SI. The times of each of the selected observations (Figs. 5–7) are also indicated on Figs. 8a–c, which show the overall times series of the aerosol optical properties ($S_p$, $\delta_p$ and $\chi'$) for each case study. For the Kasatochi and Sarychev layers (Fig. 5 and Fig. 6, respectively), the lidar ratio is relatively constant throughout the strongly backscattering regions of the stratospheric layers. The AIRS SO$_2$ signals also collocate well with these aerosols, suggesting that they are largely composed of sulfates. The curtain-average value of the lidar ratio for the two sulfate-rich layers are also very similar ($\overline{S_p} \sim 53$ sr), but

lower than the median values of the corresponding lidar ratio distributions ($\sim$60 sr; Figs. 2a, b). The Kasatochi observation corresponds to an aerosol layer that had resided in the stratosphere for $\sim$7 days whereas the Sarychev observation corresponds to a layer approximately twice the age ($\sim$14 days) of the Kasatochi layer. The mean particulate and volume depolarization ratios for the sulfate-rich layers are both relatively low ($\overline{\delta_p}$, $\overline{\delta_v} \sim 0.05$–0.10) indicating that these layers are dominated by spherical particles. The curtain-mean attenuated color ratio for the Kasatochi observation ($\overline{\chi'} = 0.37$; Fig. 5) was higher than

the Sarychev observation ($\overline{\chi'} = 0.33$; Fig. 6) although both were smaller than the Puyehue observation ($\overline{\chi'} = 0.54$; Fig. 7) indicating that the sulfate-rich layers were composed of smaller particles than the ash-rich layers.

The Puyehue layers (Fig. 7) are quite similar to the sulfate-rich layers in terms of the geometric thickness; however, the curtain-mean particulate depolarization ratio ($\overline{\delta_p}$ = 0.32), along with the AIRS ash signal, unambiguously identify this layer as being optically dominated by non-spherical ash particles. The variability in the lidar ratio for the Puyehue observation generally increases as features become more tenuous, reflecting an increase in sensitivity in the lidar ratio retrieval for transmissive layers (as discussed in Sect. 5.3). The lidar ratios are also more variable than the sulfate ratios, which may be an indication of greater inhomogeneity in the Puyehue layer observations. The curtain-mean lidar ratios for the Puyehue observation are also quite high ∼68 sr and we note that this may be due to the age of the layers (∼17 days; discussed in more detail in Sect. 6.2).

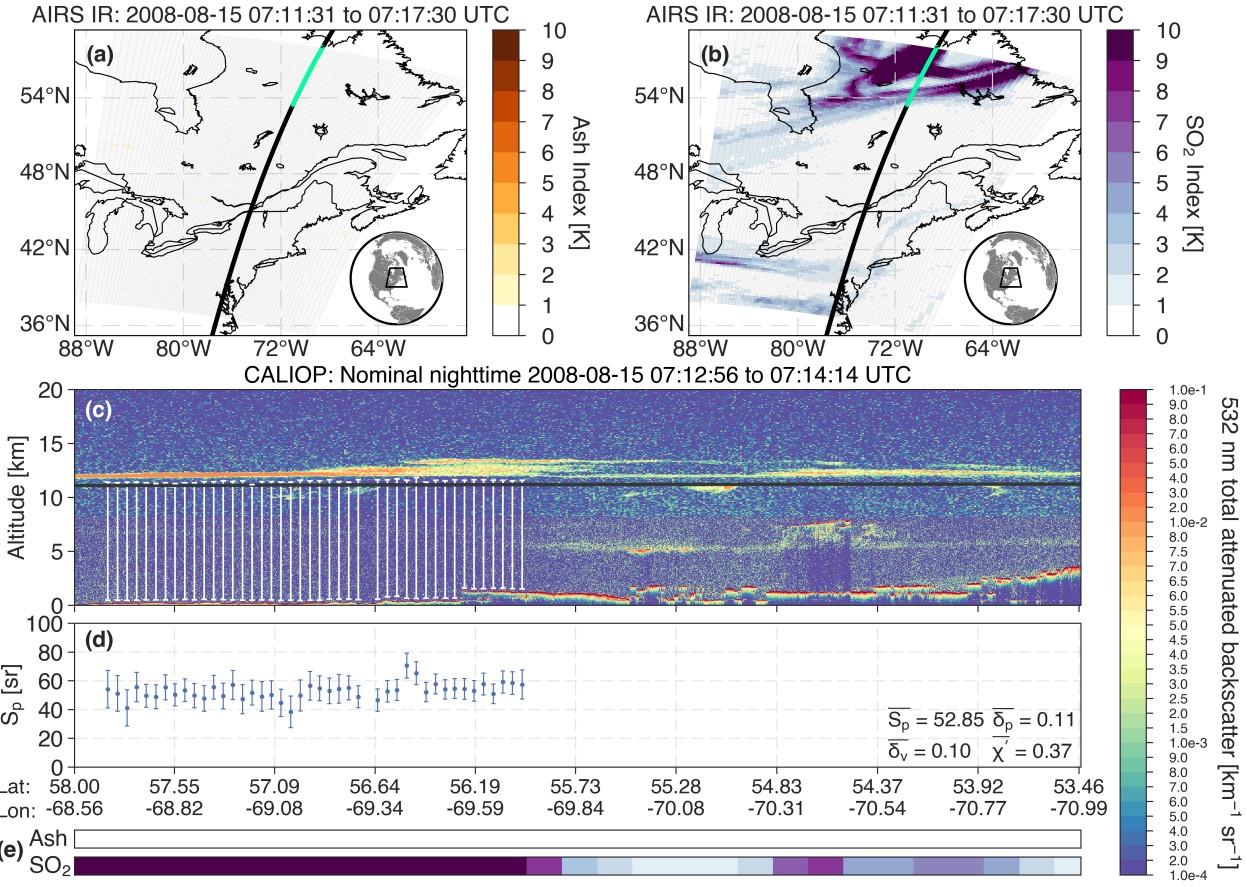

**Figure 5.** CALIOP/AIRS observations of a stratospheric volcanic sulfate-rich layer produced by the 2008 Kasatochi eruption. (a) AIRS swath with the AI (Ash Index) plotted. The CALIOP trace (black line) is over-plotted and the section of the CALIOP trace corresponding to the CALIOP curtain panel, plotted in (c), is highlighted in green. (b) Same as (a) but for the SI (SO₂ Index). (c) CALIOP curtain plot (latitude/longitude vs. total attenuated backscatter) with the GMAO tropopause height over-plotted in black and clear air analysis depths over-plotted in white. (d) Particulate lidar lidar ratio retrievals (error bars are calculated from Eq. (19)). The curtain-mean values of the particulate lidar ratio ($\overline{S_p}$), layer-integrated particulate depolarization ratio ($\overline{\delta_p}$), volume depolarization ratio ($\overline{\delta_v}$) and attenuated color ratio ($\overline{\chi'}$) are annotated on the right-hand side of the plot. (e) AI and SI AIRS pixels that have been collocated along the CALIOP track.

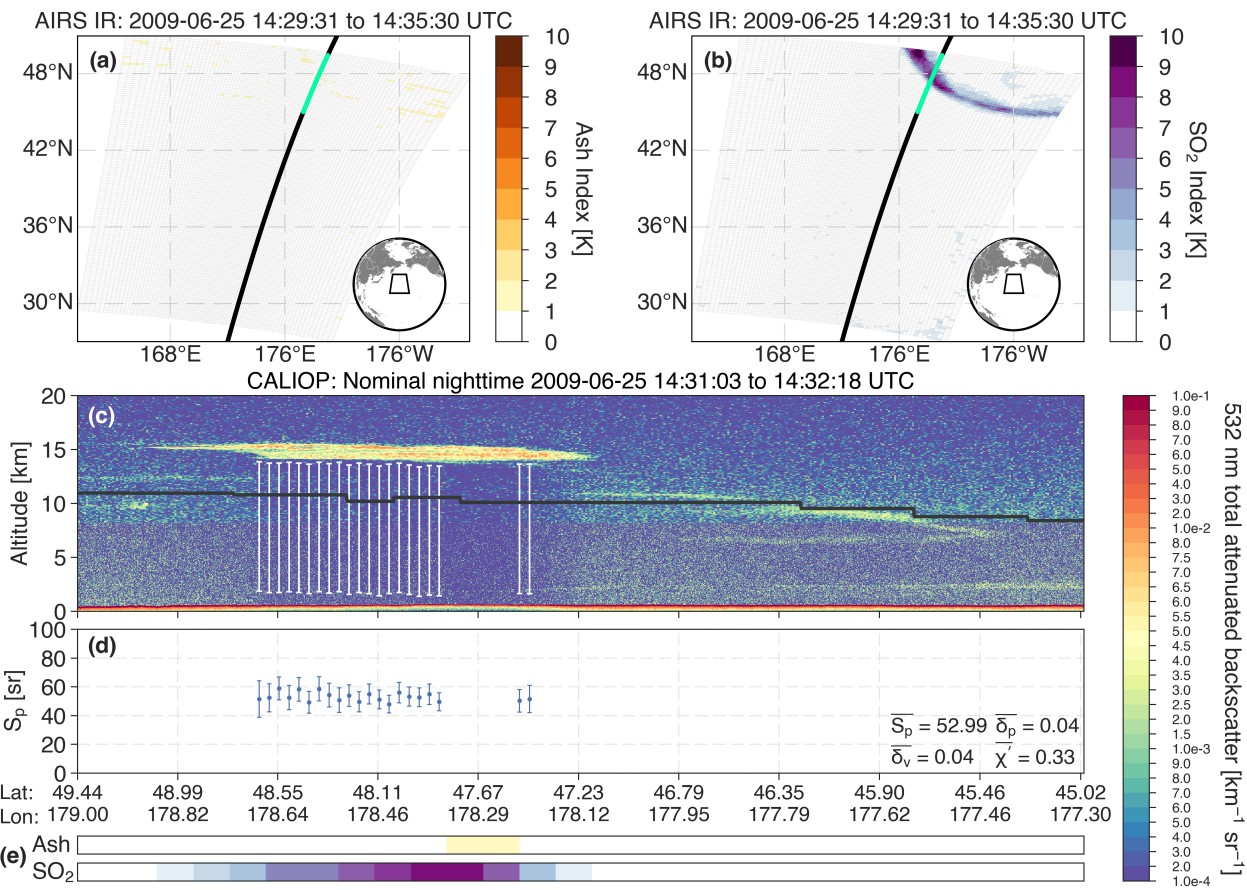

**Figure 6.** Same as Fig. 5 but for a stratospheric volcanic sulfate-rich layer produced by the 2009 Sarychev eruption.

## 6.2 Time evolution of volcanic aerosol optical properties

As volcanic aerosol layers evolve and disperse into the atmosphere their microphysical properties are expected to change with time. The Kasatochi and Puyehue layers were observable for a duration of ~12 days, while the Sarychev observations covered a time period of ~17 days. Figures 8a–c show that all observations were made more than three days after eruption onset. The Kasatochi and Puyehue volcanic aerosols were observed for a similar time period (~12 days); however, for the Puyehue case study, the aerosol layers had resided in the stratosphere for more than 11 days before the measurement period began. The Sarychev case study covered the longest observational time period, providing observations of sulfate-rich aerosols for over two weeks. All volcanic aerosol layers were subject to long-range transport across the globe as shown by the spatial distribution of observations plotted in Figs. 8j–l. The particulate lidar ratios for all three case studies were quite variable with time (Figs. 8a–c). Over these timescales (1–2 weeks) it is likely that the volcanic aerosol layers are mixing with ambient aerosol, resulting

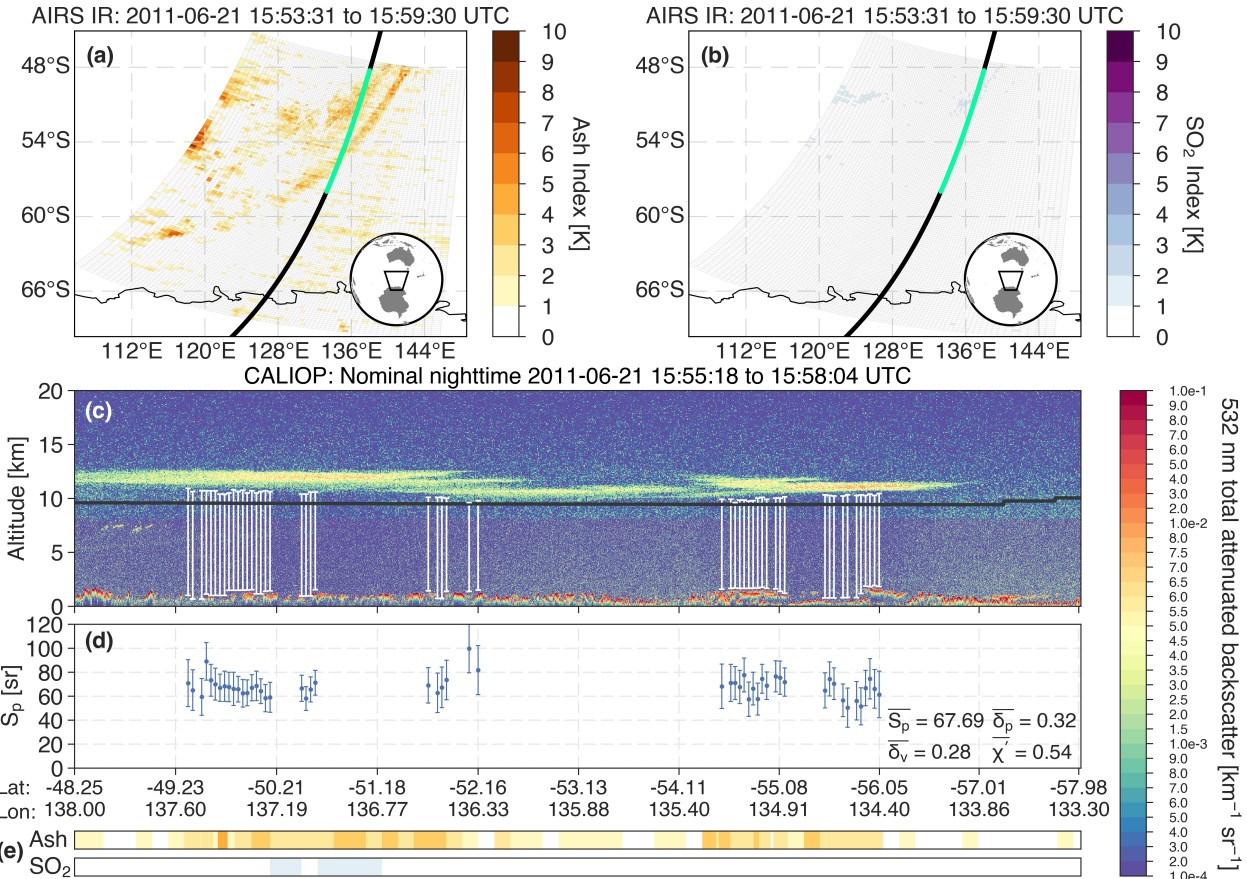

**Figure 7.** Same as Fig. 5 but for a stratospheric ash-rich volcanic aerosol layer produced by the 2011 Puyehue eruption.

in fluctuations in the lidar ratio with time. Changes in the lidar ratio may also be a result of sampling different parts of an inhomogeneous aerosol cloud.

The Puyehue lidar ratios (65–70 sr) are relatively high in comparison to previously reported volcanic ash lidar ratios (40–60 sr; Ansmann et al., 2010; Groß et al., 2012). In fact, the Puyehue lidar ratios share interesting similarities with long-range

5   transported Saharan desert dust lidar ratios (40–75 sr; Mattis et al., 2002). Mattis et al. (2002) provide two main reasons for high lidar ratios of long-range transported dust particles. The first is an increase in the fine to coarse mode particle ratio due to gravitational settling of coarse mode (diameters >1 μm) particles. The second is a large reduction in backscattering efficiency due to the non-sphericity of the particles. Both explanations are consistent with the Puyehue observations. The ash-rich aerosol layers were observed after 11 days of long-range transport (providing the necessary time for coarse mode particles to fall out)

10   and the layers were also dominated by irregular, highly depolarising ($\delta_p > 0.30$) particles.

The particulate depolarization ratios of the Puyehue layers were generally higher than the Kasatochi and Sarychev layers (Figs. 8d–i). Winker and Osborn (1992b) report similar depolarization ratios ($\delta_p \sim 0.30$) for aged (~27 days), stratospheric

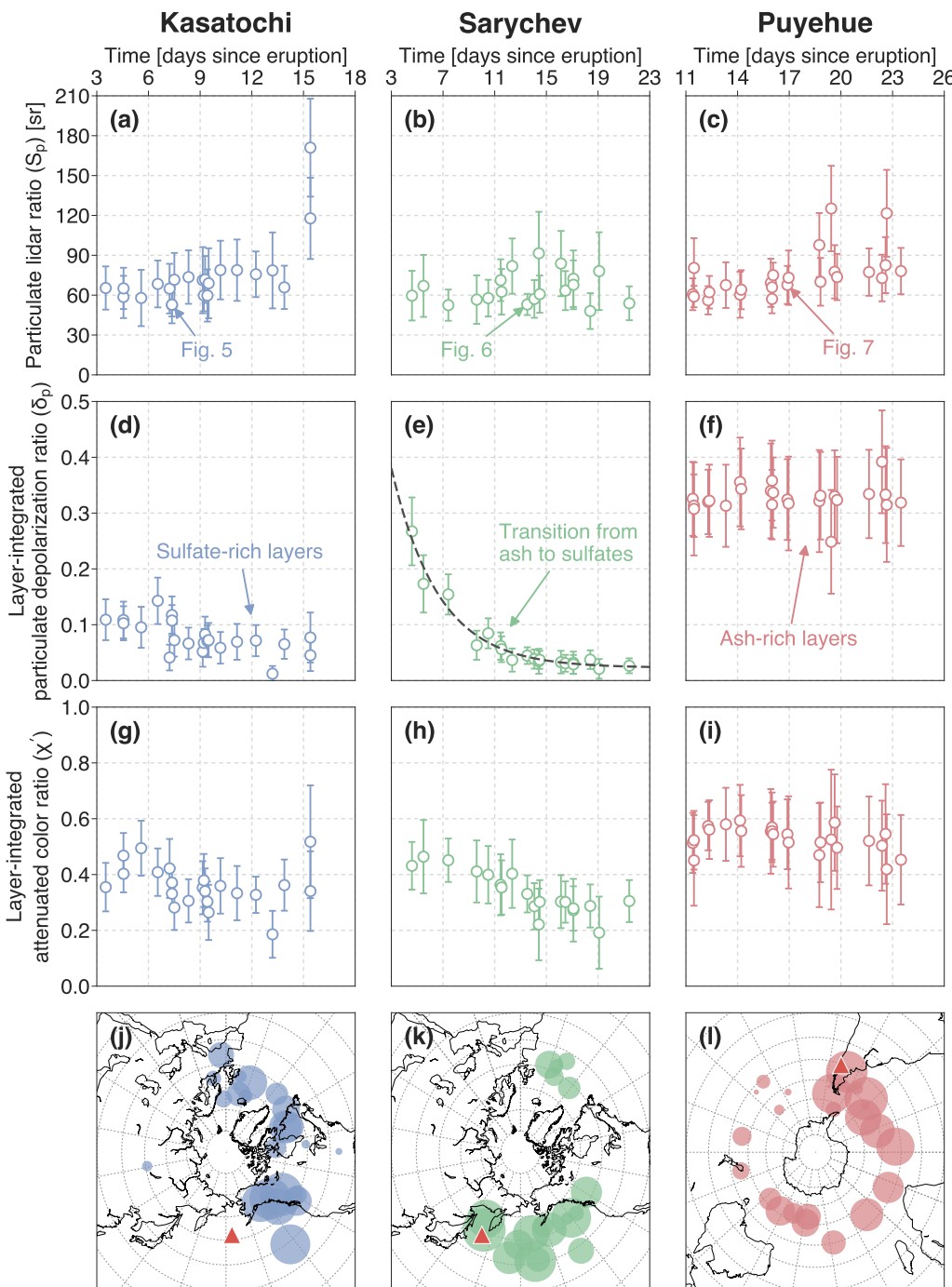

**Figure 8.** Time evolution of the optical properties for Kasatochi (left column), Sarychev (middle column) and Puyehue (right column). (a)–(c) Left axis corresponds to CALIOP curtain mean and root mean squared error (error bars) of $S_p$. (d)–(f) The same as (a)–(c) but for the layer-integrated particulate depolarization ratio ($\delta_p$). Also plotted, on (e), is an exponential fit (black dashed line) corresponding to an $e$-folding time of 3.6 days. (g)–(i) The same as (a)–(c) but for the attenuated color ratio ($\chi'$). (j)–(l) Geographic representation of the data plotted on panels (a)–(i) where the size of the data points are negatively proportional to the residence time of aerosols. Locations of volcanoes are plotted as red triangles.

($\sim$22 km) volcanic aerosol layers produced by the 1991 Pinatubo eruption. Ansmann et al. (2010), Groß et al. (2012) and Wiegner et al. (2012) report even higher particulate depolarization ratios from 0.35–0.40 for Eyjafjallajökull ash observed over Germany; however, these were observations of young (1–3 days old) tropospheric ash layers.

Over the $\sim$2.5 weeks of Sarychev CALIOP observations, $\delta_p$ is seen to decay from 0.27 to 0.03 exponentially with time. A decrease in $\chi'$ is also observed (Fig. 8h). The decay in $\delta_p$ corresponds to an $e$-folding time of 3.6 days (dashed line; Fig 8e) and may indicate that ash particles were being removed from the atmosphere during the measurement period for the Sarychev case study. Since the Sarychev layers were only analysed if the CALIOP observations were collocated with an AI < 1 K and SI $\geq$ 1 K, it is possible that the CALIOP instrument is detecting ash particles with a very weak reverse absorption signature that have not been removed by the AI threshold criterion.

Figure 8h demonstrates that $\chi'$ also decreased with time over the measurement period. Changes in $\chi'$ can be due to changes in the size, complex refractive index and shape of the aerosols being measured. It is difficult to infer, quantitatively, what the volcanic aerosol particle sizes are without assuming more about the complex refractive index and size distribution of the particles; however, we note that O'Neill et al. (2012) report effective radii of 0.25 μm for the Sarychev aerosols over the Arctic. As the attenuated color ratio is constructed based on two measurements (532 and 1064 nm attenuated backscatter) we can only use it to infer relative changes in particle size. We speculate that ash particles were present in the initial observations of the CALIOP measurements and so a combination of the sedimentation (contributing to a reduction in particle size) and sulfate formation (contributing to a change in the imaginary part of the refractive index) led to a decrease in $\chi'$ with time. Overall, the Puyehue color ratios reported here ($\chi' = 0.54 \pm 0.07$) are in agreement with the values reported by Vernier et al. (2013). These color ratios are at the low end of values reported for the free-tropospheric ash layers produced by Eyjafjallajökull (0.47–0.77; Winker et al., 2012) and considering the high particulate lidar ratios ($S_p \sim 70$ sr) and particulate depolarization ratios ($\delta_p = 0.33 \pm 0.03$) these results suggest that the CALIOP observations of the Puyehue aerosol layers are representative of layers dominated by fine mode, ash particles. The Kasatochi ($\chi' = 0.35 \pm 0.07$) and Sarychev ($\chi' = 0.32 \pm 0.07$) color ratios were, on average, quite similar but both were lower than those found for the Puyehue case study. This indicates that the Puyehue aerosol layers were composed of particles that were larger than those in the Kasatochi and Sarychev aerosol layers. The Kasatochi and Sarychev color ratios ($\chi' \sim 0.32$–0.35) were also lower than typical color ratios for desert dust ($\chi' \sim 0.45$; Liu et al., 2009), while the Puyehue color ratios ($\chi' \sim 0.53$) were generally higher. Both classes of volcanic aerosols had smaller color ratios than those CALIOP typically observes for ice ($\chi' = 0.7$–1.2) and water clouds ($\chi' = 1$–1.4; Hu et al., 2009).

## 6.3 Discriminating properties of CALIOP layer-products

Figure 9a compares the optical properties of the Kasatochi and Sarychev sulfate-rich aerosols with the Puyehue ash-rich aerosols. When combined, the volume depolarization ratios and attenuated color ratios emphasise distinctive differences between the two classes of volcanic aerosol. These optical properties are relevant to the new stratospheric aerosol classification scheme that considers $\delta_v$, $\chi'$ and $\gamma'_p$ (Tackett et al., 2016). The results of the present analysis support a sub-classification scheme, also suggested by O'Neill et al. (2012) that categorises stratospheric sulfate layers having volume depolarization ratios of $0 < \delta_v \leq 0.2$ (Fig. 9a; dashed line). Further classification could potentially be achieved using the color ratios (e.g.

$\chi' \leq 0.4$ = sulfates, $0.4 < \chi' \leq 0.7$ = ash). However, based on the aerosol layers under examination here, distinctions between ash-rich and sulfate-rich layers using $\chi'$ are less clear than distinctions made with $\delta_v$. We point out that our suggested $\delta_v$ threshold of 0.2 has been optimised for the eruption case studies considered here and that a slightly different threshold might be found for a different or larger data set. For example, Tackett et al. (2016) found a slightly lower threshold of $\delta_v = 0.15$ for the cases they examined. We also note that, for the depolarization ratio range $0.075 < \delta_v \leq 0.15$, Tackett et al. (2016) use $\chi'$ $< 0.5$ to identify stratospheric smoke. As volcanic aerosols are often composed of a complex mixture of both ash and sulfate, which changes with time, strict classification using a single threshold is challenging. In the case of ambiguous depolarization ratios ($\delta_v \sim 0.2$), supplementary information from collocated AIRS measurements may provide more insight into the likely composition of stratospheric volcanic aerosol layers.

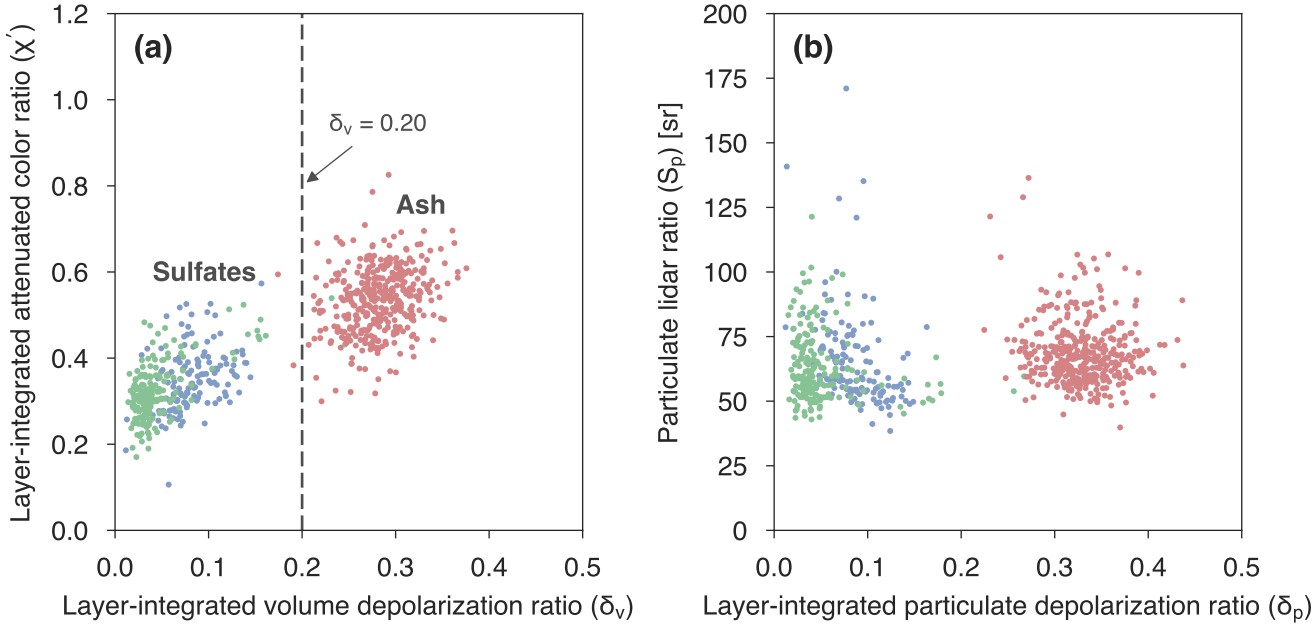

**Figure 9.** Optical properties of the Kasatochi (blue), Sarychev (green) and Puyehue (red) volcanic aerosols. (a) The relationship between the layer-integrated volume depolarization ratio and the layer-integrated attenuated color ratio. (b) The relationship between the particulate lidar ratio and the layer-integrated volume depolarization ratio.

Figure 9b shows the relationship between the particulate lidar ratio and the particulate depolarization ratio. As previously noted, the particulate lidar ratios for the Puyehue ash-rich aerosol layers and the sulfate-rich layers of Kasatochi and Sarychev were similar. This would make it difficult to discriminate between a volcanic layer dominated by ash versus a volcanic layer dominated by sulfate using $S_p$ alone. Nevertheless, these lidar ratio retrievals provide important information for distinguishing volcanic aerosols from water ($S_p \approx 20$ sr) and ice ($S_p \approx 25$ sr) clouds and could potentially be utilised in new lidar aerosol classification schemes (e.g. Groß et al., 2014).

### 6.4 Deriving an optical depth times series

In cases where the lidar ratio cannot be retrieved directly, the CALIPSO extinction retrieval (Young and Vaughan, 2009) relies on a predefined lidar ratio that is associated with a predefined type. Classification of volcanic aerosols into ash-rich and sulfate-rich layers is therefore important as the lidar ratio may change depending on the composition of the layers. The depolarization ratio appears to be the most appropriate parameter for determining whether a stratospheric volcanic layer is sulfate-rich or ash-rich. As we have shown, the lidar ratio varied with time for the case studies presented here and so the assumption of a constant lidar ratio will likely introduce errors in the retrieval of extinction profiles. Optimum results for a volcanic aerosol optical depth time series could be obtained by following the method presented here and only accepting cases where an extinction retrieval was constrained by an estimate of the two-way transmittance (i.e. extinction quality control flag equal to 1). This would most likely restrict observations to nighttime measurements of layers with optical depths > 0.2 (Fig. 4). In cases where the two-way transmittance method fails, a predefined lidar ratio would have to be used. One could use the the PDFs presented in Fig. 2 to constrain the choice of the lidar ratio. As the PDFs for the lidar ratios are positively skewed, the median lidar ratio would be best suited for this approach. For example, 60 sr for sulfate-rich ($\delta_v < 0.2$) and 67 sr for ash-rich ($\delta_v > 0.2$) layers.

### 6.5 Choice of the multiple scattering factor

In order to facilitate interpretation of the results presented in Sect. 4, $\eta$ was held constant for each case study. However, since the 'true' value of $\eta$ for volcanic aerosols is unknown we provide $S_p$ calculated for a range of different $\eta$ values in Table 3. The relationship between $\eta$ and $S_p$ for the three case studies is also shown in Fig. 10. As expected from Eq. (15), the mean particulate lidar ratio decreased as the assumed multiple scattering factor was increased.

Previously reported values of the lidar ratio (at 532 nm) provide insight into the likely range of $S_p$ for case studies considered here. The reported lidar ratios (at 532 nm) for Kasatochi and Sarychev range from 40–65 sr (Mattis et al., 2010). Although it is difficult to make direct comparisons (due to a lack of coincident observations), these values support a choice of $\eta$ closer to unity for sulfate-rich aerosols.

To our knowledge there have been no lidar ratio observations reported in the scientific literature for the Puyehue volcanic aerosols. However, ground-based lidar observations were made at Lauder, New Zealand. Nakamae et al. (2014) applied the Fernald (1984) algorithm to ground-based lidar measurements to derive aerosol (particulate) extinction profiles. They assumed a lidar ratio of 50 sr, but noted better agreement with independently derived optical depths when they set $S_p$ to 60 sr. Their initial choice of lidar ratio was based on previous reports of the lidar ratio for the Eyjafjallajökull ash layers. According to Fig. 10, a lidar ratio of 60 sr corresponds to a multiple scattering factor close to unity.

The impact of multiple scattering on CALIOP measurements can also be indicated by high depolarization ratios. Liu et al. (2011) found that effective lidar ratios ($S^* = \eta S_p$), derived from CALIOP measurements of opaque African dust layers, decrease as the volume depolarization ratio increases, an effect they ascribe to the impact of multiple scattering in denser layers. For layers with optical depths greater than 3, they found that volume depolarization ratios increased from a value of ∼0.3, typical for African dust, to ∼0.36, while the effective lidar ratios decreased to 30.5 sr from a typical value of 40 sr, implying a

**Table 3.** Mean, median and standard deviation of the particulate lidar ratio for different values of the multiple scattering factor for the Kasatochi, Sarychev and Puyehue case studies.

| Multiple scattering factor, $\eta$ | Kasatochi $S_p$ (sr) | | | Sarychev $S_p$ (sr) | | | Puyehue $S_p$ (sr) | | |
|---|---|---|---|---|---|---|---|---|---|
| | Mean | Median | Std. Dev. | Mean | Median | Std. Dev. | Mean | Median | Std. Dev. |
| 0.50 | 121.53 | 113.46 | 26.91 | 119.71 | 112.01 | 25.83 | 124.05 | 120.36 | 22.77 |
| 0.55 | 112.32 | 103.21 | 28.67 | 108.83 | 101.83 | 23.48 | 112.77 | 109.42 | 20.7 |
| 0.60 | 102.96 | 94.61 | 26.28 | 99.76 | 93.35 | 21.52 | 103.37 | 100.3 | 18.98 |
| 0.65 | 96.15 | 87.42 | 27.47 | 92.08 | 86.17 | 19.87 | 95.42 | 92.59 | 17.52 |
| 0.70 | 89.28 | 81.17 | 25.51 | 85.51 | 80.01 | 18.45 | 88.6 | 85.97 | 16.27 |
| 0.75 | 83.33 | 75.76 | 23.81 | 79.81 | 74.68 | 17.22 | 82.7 | 80.24 | 15.18 |
| 0.80 | 78.12 | 71.03 | 22.32 | 74.82 | 70.01 | 16.14 | 77.53 | 75.23 | 14.23 |
| 0.85 | 73.52 | 66.85 | 21.0 | 70.42 | 65.89 | 15.19 | 72.97 | 70.8 | 13.4 |
| 0.90 | 69.44 | 63.13 | 19.84 | 66.51 | 62.23 | 14.35 | 68.91 | 66.87 | 12.65 |
| 0.95 | 65.78 | 59.81 | 18.79 | 63.01 | 58.96 | 13.59 | 65.29 | 63.35 | 11.99 |
| 1.00 | 62.49 | 56.82 | 17.85 | 59.86 | 56.01 | 12.91 | 62.02 | 60.18 | 11.39 |

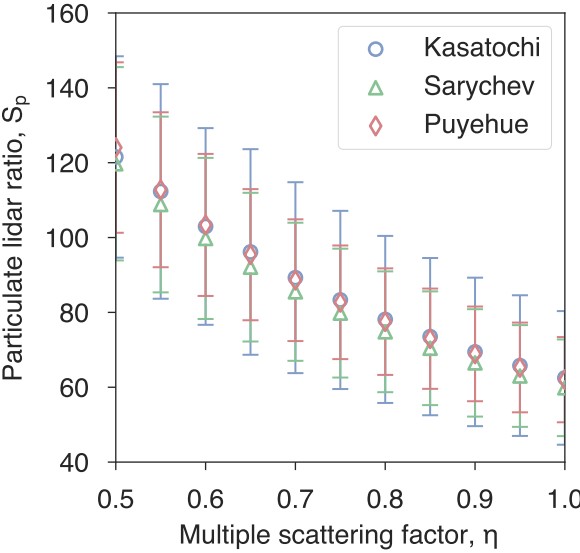

**Figure 10.** Mean particulate lidar ratios ($S_p$) for Kasatochi, Sarychev and Puyehue as a function of the multiple scattering factor, $\eta$. Error bars represent the standard deviation of $S_p$ for each case study.

multiple scattering factor of ∼0.75. For low to moderately dense layers, they found multiple scattering to be negligible. Since

the volcanic aerosol layers in this study were generally optically thin ($\tau_e < 0.8$, Fig. 4), multiple scattering effects are also expected to be small, consistent with our assumption of $\eta = 0.90$–$0.95$ for the ash-rich volcanic layers considered here.

# 7  Conclusions

By applying a two-way transmittance constraint to nighttime CALIOP observations, the equations of Fernald et al. (1972) were used to derive particulate lidar ratios ($S_p$) for two classes of volcanic aerosols (fine ash and sulfates). The combination
of CALIOP and AIRS measurements has permitted the identification and characterisation of numerous stratospheric volcanic aerosol layers produced by three recent eruptions. The median lidar ratios of the Kasatochi and Sarychev aerosols were found to be 60 sr (mean $66 \pm 19$ sr) and 59 sr (mean $63 \pm 14$ sr), respectively. The median lidar ratios are higher than the sulfate/other lidar ratio of 50 sr to be used in the new, version-4, stratospheric aerosol scheme. Further, the median lidar ratios of the aged, fine-mode ash-rich layers produced by Puyehue were found to be significantly higher (67 sr; mean $69 \pm 13$ sr) than the value
of 44 sr to be used for volcanic ash. This discrepancy suggests that ash layers could potentially be considered as two subtypes: fine (67 sr) and coarse (44 sr) mode ash.

Errors in the lidar ratio retrieval were most sensitive to errors in the effective two-way particulate transmittance constraint ($T_e^2$) when layers were optically thin. However, as $T_e^2$ approaches zero, the error in $S_p$ is limited to the error in the multiple scattering factor ($\eta$) and normalised attenuated backscatter profile ($\beta'_N(r)$). Considering the three main sources of error in the
lidar ratio retrieval ($\epsilon(\beta'_N)$, $\epsilon(T_e^2)$ and $\epsilon(\eta)$), a relative error of up to 40% is expected for the particulate lidar ratio retrievals presented here (Fig. 4).

CALIOP's stratospheric aerosol retrievals use a two-way transmittance constraint where one is available, but it is expected that the retrievals of the extinction profiles of stratospheric volcanic aerosols could be improved by setting $\eta$ to a value closer to unity. While 0.6 is a good approximation for cirrus layers (Garnier et al., 2015), it is probably an underestimate for most
stratospheric volcanic layers, which tend to have low to moderate optical depths. An underestimate of the multiple scattering factor translates to an overestimate in the particulate lidar ratio (Fig. 10) in constrained retrievals, which attempt to match the retrieved and measured two-way particulate transmittances. The use of an overestimated lidar ratio would then cause the calculated particulate extinction and optical depths to be overestimated. Determination of appropriate values for the multiple scattering factor for volcanic aerosols would further improve the accuracy of CALIOP derived lidar ratios. This could be
achieved by comparing visible and infrared optical depth retrievals (e.g. Platt, 1973; Lamquin et al., 2008; Josset et al., 2012; Garnier et al., 2015).

Several differences in the optical properties of the sulfate-rich aerosol layers versus ash-rich layers were identified through the analysis of layer-integrated optical properties. The low mean layer-integrated volume ($\delta_v$) and particulate ($\delta_p$) depolarization ratios found for the Kasatochi and Sarychev layers indicate that the assumption of collocated $SO_2$ and $SO_4^{2-}$, used
to identify sulfate-rich layers, appears to be effective and well-founded for the case studies considered. It was also shown that $\delta_v$ can be used to discriminate sulfate-rich aerosol layers from ash-rich aerosol layers, and when supplemented with the

layer-integrated attenuated colour ratio ($\chi'$) these optical properties provide useful information for new stratospheric aerosol classification schemes.

The time evolution of volcanic aerosol optical properties was also investigated. The $\delta_p$ values were consistently low ($\leq 0.10$) for the Kasatochi sulfate-rich layers and consistently high ($\geq 0.30$) for the Puyehue ash-rich layers. This suggested little change in layer composition over the measurement period for the Kasatochi and Puyehue case studies. In contrast, an exponential decay ($e$-folding time of 3.6 days) in $\delta_p$ from 0.27 to 0.03 was observed in the Sarychev layers. A transition from non-spherical to spherical aerosol particles suggested that CALIOP may have captured the formation of sulfate particles as larger irregular particles (ash) were removed. This behaviour was also characterised by a decrease in the layer-integrated attenuated colour ratio ($\chi'$) with time.

*Acknowledgements.* The authors would like to acknowledge Monash University for supporting this research through the Post-graduate Publication Award (PPA). The CALIPSO and AIRS teams are thanked for the provision of the data used in this study. The CALIPSO data were obtained from the NASA Langley Research Center Atmospheric Science Data Center. We also thank Dr Zhaoyan Liu for helpful comments on the manuscript and four anonymous reviewers whose comments helped to significantly improve the manuscript.

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
