# Peer review of "Lidar ratios of stratospheric volcanic ash and sulfate aerosols retrieved from CALIOP measurements"

_Atmospheric Chemistry and Physics, 2016_

## Referee Comment (RC1) · Anonymous Referee #3 · 27 Feb 2017

I find this paper to be an interesting contribution with a few minor issues. My main concern is with section 6.2 which I think needs some more care in how they infer things from the data. A common issue throughout the paper is that the meanings of things like color ratio and depolarization ratio are given much context (what does a value of X really mean).

Page 7, line 26. Does mean there was effectively no change in the values during measurement period?

Page 7, line 28. You commonly refer to layers as either sulfate or ash. While sometimes these layers separate themselves, other times they can be mixed in a complex fashion. You may wish to define your layers as 'layers optically dominated by ash or

sulfate aerosol' rather than imply that they are distinctly one or the other. Is it possible that complex mixing is responsible for the rather large variations in the backscatter to extinction ratio? Alternatively, is it consistent with noise or variability in the sulfate and/or ash itself?

Page 8, Some of these figures are much too small to see much detail in. I know I can blow them up to see them but my experience is that ACP makes them into JPGs for the final figures and they are always 'infinite' resolution like some bad TV show.

Page 9, Here I will be a curmudgeon, I hate VEI. People use it like it is a quantitative assessment of volcanic explosivity and I think it is disappointingly far short of that and often is not relevant to stratospheric impact. Check this out (a commercial site but the definition is correct) http://geology.com/stories/13/volcanic-explosivity-index/ . The definition is a mess.

Page 9, How do you avoid ice-rich layers? (line 20) Also, since there is a composition change from sulfate to ash, how sure are you that the changes in the color ratio are due solely to size rather than simply that they are a different color?

Page 13, line 11. I think it would be more proper to say 'unambiguously identifying this layer as containing non-spherical particles. It is not necessarily an either/or situation. . .

Section 6.2. I find much of this discussion to be speculative and perhaps the authors are over analyzing their results. Certainly, changes over time that are small compared to the measurement uncertainty is not terribly convincing. They authors are seem to forget that they never measure the same aerosol and that for an inhomogeneous cloud they cannot really be sure that some of the differences are not just variability in the cloud. The authors also do not mention that the aerosol is mixing with ambient aerosol throughout this period and so some changes are may be a result to that process. I would not bother with the humidity explanation and the comparisons with the Icelandic eruption are not likely to be particularly relevant. It is extremely common for sulfate aerosol to contain volcanic material (and meteoritic, etc.)  while optically suggesting

spherical particles. Given the high number densities after an eruption some coagulation between ash and sulfate is bound to occur in mixed layers. Perhaps some of these arguments would hold together if we had any idea of how big the ash particles are (i.e., what does the color ratio mean?). (For that matter how good do the authors believe the color ratios are? My impression of the 1064 nm channel on CALIOP is that it is not very robust though differences are real even if not correct).

---

## Referee Comment (RC2) · Anonymous Referee #4 · 28 Feb 2017

Review of "Lidar ratios of stratospheric volcanic ash and sulfate aerosols retrieved from CALIOP measurements" by Prata et al. (2017).

Volcanic aerosol optical depth from satellites are used in numerical simulations, including those presented in the Intergovernmental Panel on Climate Change reports, to assess the impact of volcanic eruptions in climate and separate natural and anthropogenic climate forcing factors. In order to derive this quantity, native backscatter measurements from CALIOP need to be converted into an extinction coefficient using a lidar ratio. The volcanic layer detection approach of this paper is based upon the combined use of AIRS and CALIOP, providing complementary information on volcanic clouds. They calculated statistical parameters associated with the optical properties

(lidar ratio, volume depolarization and attenuated color ratio) of three volcanic plumes (Sarychev, Kasatochi and Cordon) based upon the CALIOP level 2 products. They provided a thoughtful assessment of these coefficients associated with a rigorous and clear analysis of the different sources of errors. This is a very well written paper on which I don't have major comments. Thus, I strongly recommend it for publication in ACP.

I have two minor comments:

1) I believe that the proposed threshold (fig 9) to separate volcanic clouds into ash-rich and sulfate-rich categories is optimized for those cases. Indeed, Vernier et al. (2015) has shown that the pdf of the particulate depolarization ratio associated with the Kelud plume observations were indeed between those of Cordon and Sarychev/Kasatochi. Thus, the classification of volcanic cloud based upon their optical properties is challenging since those properties evolve with time depending of the presence of ash and sulfate which can also be mixed. Overall, because volcanic plumes are a mixture of two types of aerosol (external and possibly internally mixed) (sulfate and ash) which evolve with time, it makes them difficult to classify them (e.g. Kelud, Tavurvur). 2) How would you propose to use the lidar ratios calculated in this paper for deriving times series of volcanic aerosol optical depth during the months following those eruption when AIRS is not sensitive enough to detect SO2 or Ash in contrast to the CALIOP lidar measurements ? I think it would be interesting to discuss how your results can be used to derive volcanic aerosol time series.

Very nice paper!

---

## Referee Comment (RC3) · Anonymous Referee #2 · 3 Mar 2017

The paper by Prata et al. presents lidar ratio of stratospheric volcanic ash and sulfate aerosols retrieved from CALIOP measurements; an important quantity for deriving aerosol properties from a backscatter lidar like CALIOP. The paper is well suited for publication in ACP after consideration of the following comments:

General comments:

The description of the used method is hardly to follow for a reader less experienced with this method. The manuscript often refers to former papers for important equations. The chapter should be reworked in a way that all important points are included in this manuscript. It should also be clearly worked out why the a priori lidar ratio which was used for the calculation of the particle backscatter coefficient and the effective two-way

transmittance does not affect the retrieved lidar ratio.

Minor comments:

The mean depolarization ratio for Puyehue in the abstract (0.28) differs from the mean value given in Table 2 (0.29).

You report about an exponential decay in the mean depolarization ratio for the Sarychev layer with time. Do you see changes also in one or more of the other properties?

Page 4, line 31: At this point '$\eta$' is not defined. Please make sure that all variables are defined when using them the first time.

Section 3.1: How is the BTD algorithm defined? Please give more information about this.

Section 3.1: Why do the conditions differ for the different volcanic layers?

Section 3.2: What is meant by the 'mean scattering ratio'?

Section 4.2: Mean color ratio for Sarychev layer does not agree with value given in Table 2.

Section 4.2: In the Abstract it was reported that the depolarization ratio exponentially decreased with time. This is not reported in Section 4.2. As the change in the optical properties is an important point and thus reported in the Abstract it should also be referred to in the description of the Sarychev layer. How do the other optical properties behave? What are the properties in the beginning of the observations (, mid) and end? This is reported quite lately in the manuscript. To which periods does the mean value correspond to? If the mean values are calculated from the whole period what is the significance of this mean value?

Section 4.3: How is a valid lidar ratio profile defined (time/length)? The number of cases/profiles resulting into the mean values should also be given for the other cases. Information about the CALIOP measurements (number, time, days, and lo-

cation) should be given for the different cases should.

Section 4.3: Standard deviation of color ratio does not agree with the value given in Table 2.

Section 5.1: What is meant by the aerosol scattering ratio?

Section 5.3: Why did you use the error calculation according to Equation 9? This formula is used for the calculation of random (statistical) errors. To my understanding, the errors considered in this manuscript are not random errors and thus the total error should be calculated from the absolute error values of the considered parameters.

Figures 5-7: Please indicate the aerosol free regions below and above the volcanic layer.

Section 6.1: The mean values of the lidar ratio for the Kasatochi and Sarychev layers shown in this case studies are smaller than the mean values reported for the whole measurements for these layers. Can you give more information about the changes over the time? Maybe give a time series of the lidar ratio for the different volcanic layers to illustrate the changes and / or variability over time. Otherwise the mean values of the case studies or the mean values over all suffer the loss of significance.

Section 6.2: It is right that the measurements of the different volcanic layers correspond to different stages / ages of the volcanic layer. However the way it is described here could lead to misinterpretation of this information, as one could think that the measurements of the three volcanic layers can be related to each other and show an alteration of volcanic aerosol layers during time.

Section 6.2: The increase of the Puyehue ash layer with time is small compared to the uncertainties of the retrieved property, thus the statement derived from this changes is very speculative.

Section 6.3, discussion about high lidar ratios for Puyehue: Should not the loss of the large particles also be reflected in the depolarization ratio? No changes are obvious

there.

Page 8, lines 7-8: This statement about the volume and the particle depolarization ratio is misleading.

---

## Referee Comment (RC4) · Anonymous Referee #1 · 18 Mar 2017

General

The paper is well written. Carefully analysed CALIOP observations are presented. The paper is appropriate for ACP.

The only negative and confusing point is that obviously the volume depolarization ratio and volume color ratio are used instead of the particle depolarization ratio and particle color ratio. But I am not sure what is shown. The authors have to clarify that when discussing equations 1 and 2, see details.

Minor revisions are at least required. However, major revisions (switch to particle depolarization ratio) would significantly improve the paper.

[Figure]

Details:

Abstract:

P1, L9: Please state the wavelength (532 nm) again in the case of the volume depolarization ratio.

P1, L10-12: A volume depolarization ratio of 0.08, 0.05, 0.25 tells us almost nothing as long as we do not know the backscatter ratio (total-to-Rayleigh backscatter). So again, why not trying to determine the particle depolarization ratio? At least for a few examples.

Introduction:

P2, L22: Later on, in this paper, you mention the Mattis paper which also deals with the same volcanic eruptions in the high northern latitudes in 2008 and 2009. I checked that paper and found lidar ratios and depolarisation ratios for 355 and 532 nm for high-northern-latitude volcanic aerosol in the upper troposphere and stratosphere.

So, I was surprized that you did not give any reference to this paper in the introduction. Is there a specific reason, or did you simply forget? Mattis found lidar ratios of 30-40sr for 532nm and 60-80 sr for 355nm in August 2008 (upper troposphere, clearly related to volcanic aerosol), and 30-50sr for both wavelength between 14-18 km height one year later. And, by the way, Mattis found volume depolarization ratios of 0.015. Such low numbers really indicate sphercial particles, in contrast to your high numbers of 0.05 to 0.08 for the volume depolarization ratio, so that I started to think about the particle depolarization ratio.

So, please give proper reference to that Mattis paper in the introduction!

Instead, you mention papers that deal with volcanic layers in the lower troposphere. Please give the heights of these volcanic layers so that the reader can make his/her own conclusion how useful such information is in a paper dealing with stratospheric volcanic layers.

P4, L19-20: Again, I am not very happy that you do not make any attempt to provide particle depolarization ratios.

Section3: I appreciate the careful consideration of potential multiple scattering effects!

Now, I got confused! Equation 1 leads, to my opinion, to the particle depolarization ratio. Right? Please clarify that! Are these cross and co-polarized backscatter coefficients for particles???? or for the total (Rayleigh plus particle) backscattering. Please make that very very clear!

If that is for the total backscatter then please put an index 'p' to the ones in equation 2!... or are these total (Rayleigh plus particle) backscatter coefficients as well???

I got confused because equation 3 deals with the Fernald 1972 approach! So, you have the potential to compute particle backscatter coefficients and particle depolarization ratios when using the later Fernald method (Appl. Opt.l, 1984). So, why not presenting particle related quantities: lidar ratio, depolarization ratio, color ratio?

Figure 2 is very nice, but I am missing the particle depolarization ratio, and obviously the color ratio is also for Rayleigh plus particle backscatter coefficients, and thus not very helpful. . ... But, at the moment, I am not sure what is shown.

All the results in the figures are nice (figures 5,6,7,8 ,9), but I am still confused to see PARTICLE lidar ratios together with information on VOLUME depolarization ratios and VOLUME color ratio.

Correlations (Fig.9)1of PARTICLE lidar ratio versus VOLUME depolarization ratio are poor!!! Apples and oranges are correlated, to my opinion.

May be it is simply not easy to compute particle depolarization ratios and particle color ratios. But at least a figure showing both, the volume and particle depolarization ratio and maybe the same for the color ratio is required to convince the reader that such correlations as in Figure 9 are useful.

---

## Author Comment (AC1) · 17 May 2017

**Response to RC1**

I find this paper to be an interesting contribution with a few minor issues. My main concern is with section 6.2 which I think needs some more care in how they infer things from the data. A common issue throughout the paper is that the meanings of things like color ratio and depolarization ratio are given much context (what does a value of X really mean).

Response:

   The authors thank the reviewer for comments on the manuscript. As suggested, Section 6.2 has been revised and more care has been taken when interpreting the time evolution of the volcanic aerosols. Upon revisiting the time series analysis we noticed an error in the code that was used to construct the time series from the lidar data. In the original code, the **cumulative** mean was being calculated for each optical property in the time series. The time of each observation was also, incorrectly, calculated as a cumulative mean, which resulted in the incorrect residence time for each data point presented in Fig. 8 of the original manuscript. In the revised manuscript, this error has been corrected so that the curtain means and root mean square errors are calculated for each CALIOP/AIRS observation and are plotted together with the curtain mean of the time of each observation. Note that we define the 'curtain mean' as the mean of all CALIOP layer optical properties (i.e. $S_p$, $\delta_v$, $\delta_p$ and $\chi'$) within a collocated AIRS granule, which equates to a ~6 minute subset of a CALIOP granule. This revision only affects the original data plotted in Fig. 8 of the original manuscript. It also impacts the calculation of the $e$-folding time of the Sarychev depolarization ratios. We have therefore attached the revised version of Fig. 8 (Figure 1 of this document) below.

   We have also more explicitly defined both the depolarization ratio and the color ratio in the revised manuscript. We note, though, that the color ratio is constructed based on only two measurements (532 and 1064 nm attenuated backscatter) and so it is difficult to infer, quantitatively, what the volcanic aerosol particle sizes are without assuming more about the complex refractive index and size distribution of the particles. It can, however, be used to infer relative size. This is explained in more detail in the revised manuscript and in the responses to comments that follow.

Page 7, line 26. Does mean there was effectively no change in the values during measurement period?

Response:

   Indeed, there was little change in the optical properties during the measurement period for the Kasatochi case study. We refer the reviewer to the revised Fig. 8 (Figure 1 below), which shows how the optical properties changed over time during the measurement time period.

Page 7, line 28. You commonly refer to layers as either sulfate or ash. While sometimes these layers separate themselves, other times they can be mixed in a complex fashion. You may wish to define your layers as layers optically dominated by ash or sulfate aerosol rather than imply that they are distinctly one or the other. Is it possible that complex mixing is responsible for the rather large variations in the backscatter to extinction ratio? Alternatively, is it consistent with noise or variability in the sulfate and/or ash itself?

Response:

   Thank you for suggesting this. In the revised manuscript, the authors are careful to refer to layers as 'ash-rich', 'sulfate-rich' and as being 'optically dominated' by either ash or sulfate. To the reviewer's second point: as we are measuring the aerosol layers over a number of days across the globe, it is possible that complex mixing of ambient aerosol will be occurring over time. Sedimentation, dehy-

[Figure]

Figure 1: Revised version of Fig. 8 of the original manuscript.

dration and coagulation processes are also likely to be occurring. Therefore some variability in the lidar ratio should be expected. We note, however, that the lidar ratio retrieval becomes more sensitive (and uncertain) to changes in the return backscatter signal as the two-way transmittance approaches 1 (see Fig. 4 of original manuscript). As the majority of the aerosol layers were optically thin ($\tau_e < 1$), the large variability (high standard deviations) in the lidar ratio PDFs (Fig. 2 of original manuscript) is probably also, in part, due to the high sensitivity to noise in the backscatter return signal. However, based on the observational evidence provided by the color and depolarization ratios, we believe that CALIOP has captured compositional changes in the volcanic aerosols under examination; particularly for the Sarychev case (see Figure 1e of this document).

Page 8, Some of these figures are much too small to see much detail in. I know I can blow them up to see them but my experience is that ACP makes them into JPGs for the final figures and they are always 'infinite' resolution like some bad TV show.

Response:
    To improve readability, we have increased the size of all figures on page 8. We have also increased the font size in all figures. The majority of our figures are in pdf (vectorised) format and so no resolution will be lost in the ACP typesetting stage for these figures. All other figures are in png (non-lossy) format with a dpi of 600. We have been careful to follow the ACP guidelines on producing high quality figures as described here: http://www.atmospheric-chemistry-and-physics.net/for_authors/manuscript_preparation.html

Page 9, Here I will be a curmudgeon, I hate VEI. People use it like it is a quantitative assessment of volcanic explosivity and I think it is disappointingly far short of that and often is not relevant to stratospheric impact. Check this out (a commercial site but the definition is correct) http://geology.com/stories/13/volcanic-explosivity-index/ . The definition is a mess.

Response:
    Reference to the VEI has been removed in the revised manuscript as the authors agree that its use here is not relevant to the study.

Page 9, How do you avoid ice-rich layers? (line 20) Also, since there is a composition change from sulfate to ash, how sure are you that the changes in the color ratio are due solely to size rather than simply that they are a different color?

Response:
    Ice-rich layers are avoided based on the Ash Index (AI) criterion. If a stratospheric layer were ice-rich then we would expect the AI to be strongly negative (Prata et al., 2015). Since our criterion is set so that we only accept AI $\geq 1$ we assume that ice-rich layers have been removed from the analysis. The explicit definition of the AI has been included in the revised manuscript.
    To the reviewer's second point, we never suggest that there is a transition from sulfate to ash, rather, we suggest that there is a composition change from ash to sulfate (based on the depolarization and color ratio changes for the Sarychev case). Sulfate aerosols are generally in the 0.1–1 $\mu$m radius size range and ash particles that have resided in the stratosphere for more than 2 weeks would likely be sub-micron to micron size. Indeed, O'Neill et al. (2012) report effective radii of 0.25 $\mu$m for the Sarychev aerosols over the Arctic. This means that the size of the particles under examination is less than/comparable to the sampling wavelengths (532 nm and 1064 nm). In this sense, we are talking about scattering and absorption, rather than reflection, which means changes in the 'color' of the particles (in the usual sense of the word) could not be inferred using CALIOP measurements.

The color ratio can change due to changes in the size, complex refractive index and shape of the aerosols being measured. We speculate that ash particles were present in the initial observations of the CALIOP measurements for Sarychev case and so a combination of the sedimentation (contributing to a reduction in particle size) and sulfate formation (contributing to a change in the imaginary part of the refractive index) led to changes in the color ratio. This discussion will be included in the revised manuscript.

Page 13, line 11. I think it would be more proper to say 'unambiguously identifying this layer as containing non-spherical particles. It is not necessarily an either/or situation...

Response:
Accepted. The sentence in the revised manuscript has been amended to read:
"The Puyehue layers (Fig. 7) are quite similar to the sulfate-dominated layers in terms of the geometric thickness; however, the layer-integrated optical properties, along with the AIRS ash signal, unambiguously identify this layer as containing non-spherical ash particles."

Section 6.2. I find much of this discussion to be speculative and perhaps the authors are over analyzing their results. Certainly, changes over time that are small compared to the measurement uncertainty is not terribly convincing. They authors are seem to forget that they never measure the same aerosol and that for an inhomogeneous cloud they cannot really be sure that some of the differences are not just variability in the cloud. The authors also do not mention that the aerosol is mixing with ambient aerosol throughout this period and so some changes are may be a result to that process. I would not bother with the humidity explanation and the comparisons with the Icelandic eruption are not likely to be particularly relevant. It is extremely common for sulfate aerosol to contain volcanic material (and meteoritic, etc.) while optically suggesting spherical particles. Given the high number densities after an eruption some coagulation between ash and sulfate is bound to occur in mixed layers. Perhaps some of these arguments would hold together if we had any idea of how big the ash particles are (i.e., what does the color ratio mean?). (For that matter how good do the authors believe the color ratios are? My impression of the 1064 nm channel on CALIOP is that it is not very robust though differences are real even if not correct).

Response:
Thank you for this comment. Section 6.2 has been revised to accomodate the reviewers suggestions. We have removed the discussion of the changes in layer-integrated attenuated backscatter and the coagulation and condensation processes as we agree that this part of Sect. 6.2 is speculative based on the evidence (see also revised Figure 1 of this document). We instead comment on the fact that changes in the lidar ratio may be due to variability and inhomogeneity in the aerosol layers. We have also now incorporated discussion on the possibility of volcanic aerosol mixing with ambient aerosol during the measurement period. The humidity explanation has been removed; however, as there are few ground-based observations of the volcanic ash lidar ratio together with depolarization ratio (at 532 nm), we believe that comparison of the Eyjafjallajökull observations with the Puyehue observations is justified.
As discussed above, while we are not able to retrieve particle size, the color ratio can indicate relative changes in particle size. For these reasons we can infer that the Puyehue particles were larger than the Kasatochi and Sarychev particles. Reference to O'Neill et al. (2012) has been included in the revised discussion as they report on particle sizes for the Sarychev case.
The quality of the layer-integrated attenuated color ratios depends on the correct identification of the layer-top and base, the reliability of the 532 and 1064 nm calibration constants and the SNR. The 1064 nm channel calibration depends on the assumption that the color ratio for high cirrus clouds is

1. The calibration procedure is described in Section 7.1.2.2 of the level 1 ATBD (Winker et al., 2006) and the assumption of the cirrus cloud color ratio was determined to be justified based on a validation study using the Cloud-Physics Lidar (Vaughan et al., 2010). We therefore believe that, while there may be some variability in the calibration of the 1064 nm channel, the color ratios used here are robust enough to infer relative changes in particle size.

**References**

O'Neill, N. T., Perro, C., Saha, A., Lesins, G., Duck, T. J., Eloranta, E. W., Nott, G. J., Hoffman, A., Karumudi, M. L., Ritter, C., Bourassa, A., Abboud, I., Carn, S. A., and Savastiouk, V.: Properties of Sarychev sulphate aerosols over the Arctic, J. Geophys. Res., 117, D04 203–21, 2012.

Prata, A. T., Siems, S. T., and Manton, M. J.: Quantification of volcanic cloud top heights and thicknesses using A-train observations for the 2008 Chaitén eruption, J. Geophys. Res. Atmos., 120, 2928–2950, 2015.

Vaughan, M. A., Liu, Z., McGill, M. J., Hu, Y., and Obland, M. D.: On the spectral dependence of backscatter from cirrus clouds: Assessing CALIOP's 1064 nm calibration assumptions using cloud physics lidar measurements, J. Geophys. Res. Atmos., 115, D14 206, 2010.

Winker, D. M., Hostetler, C. A., Vaughan, M. A., and Omar, A. H.: CALIOP Algorithm Theoretical Basis Document, Part 1: CALIOP Instrument, and Algorithms Overview, Release, 2006.

---

## Author Comment (AC2) · 17 May 2017

**Response to RC2**

*In the following response, reviewer comments are in black and author responses are in blue.*

Review of Lidar ratios of stratospheric volcanic ash and sulfate aerosols retrieved from CALIOP measurements by Prata et al. (2017).

Volcanic aerosol optical depth from satellites are used in numerical simulations, including those presented in the Intergovernmental Panel on Climate Change reports, to assess the impact of volcanic eruptions in climate and separate natural and anthropogenic climate forcing factors. In order to derive this quantity, native backscatter measurements from CALIOP need to be converted into an extinction coefficient using a lidar ratio. The volcanic layer detection approach of this paper is based upon the combined use of AIRS and CALIOP, providing complementary information on volcanic clouds. They calculated statistical parameters associated with the optical properties (lidar ratio, volume depolarization and attenuated color ratio) of three volcanic plumes (Sarychev, Kasatochi and Cordon) based upon the CALIOP level 2 products. They provided a thoughtful assessment of these coefficients associated with a rigorous and clear analysis of the different sources of errors. This is a very well written paper on which I dont have major comments. Thus, I strongly recommend it for publication in ACP.

I have two minor comments:
1) I believe that the proposed threshold (fig 9) to separate volcanic clouds into ash-rich and sulfate-rich categories is optimized for those cases. Indeed, Vernier et al. (2015) has shown that the pdf of the particulate depolarization ratio associated with the Kelud plume observations were indeed between those of Cordon and Sarychev/Kasatochi. Thus, the classification of volcanic cloud based upon their optical properties is challenging since those properties evolve with time depending of the presence of ash and sulfate which can also be mixed. Overall, because volcanic plumes are a mixture of two types of aerosol (external and possibly internally mixed) (sulfate and ash) which evolve with time, it makes them difficult to classify them (e.g. Kelud, Tavurvur). 2) How would you propose to use the lidar ratios calculated in this paper for deriving times series of volcanic aerosol optical depth during the months following those eruption when AIRS is not sensitive enough to detect SO2 or Ash in contrast to the CALIOP lidar measurements? I think it would be interesting to discuss how your results can be used to derive volcanic aerosol time series.
Very nice paper!

Response:
    The authors thank the reviewer for their thoughtful comments on the manuscript. Response to 1) The authors agree that the proposed threshold is optimised for the case studies considered. However, as we attempted to separate ash-rich layers (AI $\geq$ 1 K and SI $<$ 1 K) and sulfate-rich layers (AI $<$ 1 K and SI $\geq$ 1 K) using AIRS, we expect that the majority of mixed layers (sulfate/ash) would exhibit an AI $>$ 1 K and SI $>$ 1 K and so would have been removed from our analysis. Our depolarization measurement results, therefore, highlight the two extreme cases (i.e. ash-rich or sulfate-rich) and so, as the reviewer has stated, values falling between the Puyehue and Kasatochi/Sarychev values would likely be a mixture of sulfate and ash (e.g. Vernier et al., 2016). Classification of volcanic aerosols into ash-rich and sulfate-rich layers is important as the lidar ratio may change depending on the composition of the layers. We approached this problem with the operational extinction retrieval in mind; when the lidar ratio cannot be retrieved directly, the aerosol must be classified as a predefined type (associated with a predefined lidar ratio). We have proposed a method for detection, using native CALIOP measurements, of sulfates and ash and have given values of the lidar ratio for

these particular case studies. We have acknowledged the reviewers point in the revised manuscript as follows:

"We also note that the suggested $\delta_v$ threshold will be optimised for the eruption case studies considered here. As volcanic aerosols are often composed of a complex mixture of both ash and sulfate, which changes with time, strict classification using a single threshold is challenging. In the case of ambiguous depolarization ratios ($\delta_v \sim 0.2$), supplementary information from collocated AIRS measurements may provide more insight into the likely composition of stratospheric volcanic aerosol layers."

Response to 2) This is the reason we explored the use of CALIOP-measured parameters for discriminating volcanic ash from sulfate after first identifying ash-rich and sulfate-rich layers using independent detection from AIRS. The depolarization ratio appears to be the most appropriate parameter for determining whether a stratospheric volcanic layer is sulfate-rich or ash-rich. As we have shown, the lidar ratio varies with time and so the assumption of a constant lidar ratio will likely introduce errors in the retrieval of extinction profiles. Optimum results for a volcanic aerosol optical depth time series would therefore be obtained by following the method presented here and only accepting cases where an extinction retrieval is constrained by an estimate of the two-way transmittance (i.e. QC_flag = 1). This would most likely restrict observations to nighttime measurements of layers with optical depths $> 0.2$ (see Fig. 4 of original manuscript). In cases where the two-way transmittance method fails, a predefined lidar ratio would have to be used. One could use the the PDFs presented in Fig. 2 of the original manuscript to constrain the choice of the lidar ratio. As the PDFs for the lidar ratios are positively skewed, we suggest using the median lidar ratios for this approach. For example, 60 sr for sulfate-rich ($\delta_v < 0.2$) and 67 sr for ash-rich ($\delta_v > 0.2$) layers. We have included this discussion as a subsection of Section 6 of the revised manuscript.

**References**

Vernier, J.-P., Fairlie, T. D., Deshler, T., Natarajan, M., Knepp, T., Foster, K., Weinhold, F. G., Bedka, K. M., Thomason, L., and Trepte, C.: In situ and space-based observations of the Kelud volcanic plume: The persistence of ash in the lower stratosphere, J. Geophys. Res. Atmos., 2016.

---

## Author Comment (AC3) · 17 May 2017

**Response to RC3**

*In the following response, reviewer comments are in black and author responses are in blue.*

The paper by Prata et al. presents lidar ratio of stratospheric volcanic ash and sulfate aerosols retrieved from CALIOP measurements; an important quantity for deriving aerosol properties from a backscatter lidar like CALIOP. The paper is well suited for publication in ACP after consideration of the following comments:

General comments:

The description of the used method is hardly to follow for a reader less experienced with this method. The manuscript often refers to former papers for important equations. The chapter should be reworked in a way that all important points are included in this manuscript. It should also be clearly worked out why the a priori lidar ratio which was used for the calculation of the particle backscatter coefficient and the effective two-way transmittance does not affect the retrieved lidar ratio.

Response:
    The authors thank the reviewer for this comment. In the revised manuscript, the methods section has been reworked to include key equations and describes the relevant steps needed to retrieve the lidar ratio using the Fernald method. Explicit definitions of the AI and SI have now also been included.
    In regard to the *a priori* lidar ratio, the reviewer has misunderstood the retrieval method here. The *a priori* lidar ratio is not used to calculate the particulate backscatter coefficient and effective two-way transmittance. This is because the particulate backscatter coefficient, $\beta_p(r)$, does not appear in the two-component lidar ratio solution (see Eq. (3) of original manuscript). Also, the effective two-way transmittance is measured based on the mean attenuated scattering ratio - and so no *a priori* assumptions of the lidar ratio are required to estimate the transmittance. For the top layer, it is measured as the mean attenuated scattering ratio in a clear air region immediately below the layer.
    The iterative lidar ratio solution in Eq. (3) (original manuscript, now Eq. (7) in revised manuscript), however, does require an initial estimate of the lidar ratio to begin the iteration. The choice of the initial lidar ratio will affect the number of iterations required for consecutive solutions to converge. As noted in Fernald et al. (1972), in general, Eq. (3) will converge rapidly but will converge more slowly for very clean atmospheres. In practice we have found that solutions converge rapidly when initialising Eq. (3) with the result of Eq. (7) (original manuscript, now Eq. (15) of revised manuscript).

Minor comments: The mean depolarization ratio for Puyehue in the abstract (0.28) differs from the mean value given in Table 2 (0.29).

Response:
    This error has now been corrected in the revised manuscript.

You report about an exponential decay in the mean depolarization ratio for the Sarychev layer with time. Do you see changes also in one or more of the other properties?

Response:
    Indeed we do see changes in both the lidar ratio and the layer-integrated attenuated color ratio with time for the Sarychev case study. The attenuated color ratio also decreases with time; similar to the depolarization ratio. The lidar ratio is quite variable showing no significant increasing or decreasing trend with time. We have now made mention of the change in color ratio with time for the Sarychev case in the abstract of the revised manuscript.

Upon revisiting the time series analysis we noticed an error in the code that was used to construct the time series from the lidar data. In the original code, the **cumulative** mean was being calculated for each optical property in the time series. The time of each observation was also, incorrectly, calculated as a cumulative mean, which resulted in the incorrect residence time for each data point presented in Fig. 8 of the original manuscript. In the revised manuscript, this error has been corrected so that the curtain means and root mean square errors are calculated for each CALIOP/AIRS observation and are plotted together with the curtain mean of the time of each observation. Note that we define the 'curtain mean' as the mean of all CALIOP layer optical properties (i.e. $S_p$, $\delta_v$, $\delta_p$ and $\chi'$) within a collocated AIRS granule, which equates to a $\sim$6 minute subset of a CALIOP granule. This revision only affects the original data plotted in Fig. 8 of the original manuscript. It also impacts the calculation of the $e$-folding time of the Sarychev depolarization ratios. We have therefore attached the revised version of Fig. 8 (Figure 1 of this document) below.

Page 4, line 31: At this point '$\eta$' is not defined. Please make sure that all variables are defined when using them the first time.

Response:

The multiple scattering factor is defined on page 4 line 18 of the original manuscript (before the line that the reviewer is referring to). However, in Sect. 2 of the revised manuscript we have been more explicit in defining $\eta$:

"We note that the effective two-way transmittance profile, $T_{e,\lambda}^2(0, r)$, is related to the particulate two-way transmittance profile via $T_{e,\lambda}^2(0, r) = T_{p,\lambda}^{2\eta}(0, r)$, where $\eta$ is defined here as the multiple scattering factor (Platt, 1973)."

Section 3.1: How is the BTD algorithm defined? Please give more information about this.

Response:

The BTD algorithms for the AI and SI are defined as

$$\text{SI} = \text{BT}(1407.2 \text{ cm}^{-1}) - \text{BT}(1371.5 \text{ cm}^{-1}). \tag{1}$$

and

$$\text{AI} = \text{BT}_1 - \text{BT}_2 + \text{BT}_3 - \text{BT}_4 \tag{2}$$

where

$$\text{BT}_1 = \frac{1}{4}[\text{BT}(856.44 \text{ cm}^{-1}) + \text{BT}(856.75 \text{ cm}^{-1}) \\ + \text{BT}(857.06 \text{ cm}^{-1}) + \text{BT}(857.37 \text{ cm}^{-1})],$$

$$\text{BT}_2 = \frac{1}{4}[\text{BT}(964.25 \text{ cm}^{-1}) + \text{BT}(965.04 \text{ cm}^{-1}) \\ + \text{BT}(965.44 \text{ cm}^{-1}) + \text{BT}(966.24 \text{ cm}^{-1})],$$

$$\text{BT}_3 = \frac{1}{2}[\text{BT}(1131.79 \text{ cm}^{-1}) + \text{BT}(1133.96 \text{ cm}^{-1})]$$

[Figure]

Figure 1: Revised version of Fig. 8 of the original manuscript.

and

$$BT_4 = \frac{1}{2}[BT(1080.92 \text{ cm}^{-1}) + BT(1082.41 \text{ cm}^{-1})].$$

Here $BT(\nu)$ is the brightness temperature measured at wavenumber, $\nu$. These equations have been added to Sect. 3.1 of the revised manuscript.

Section 3.1: Why do the conditions differ for the different volcanic layers?

Response:
    We assume that the reviewer is referring to the SI and AI threshold conditions. The reason the conditions differ is that we are looking for a volcanic ash signal for the Puyehue case study and an $SO_2$ signal for the Kasatochi and Sarychev case studies. In order to detect volcanic ash we require that the AI be greater than or equal to 1 K and the SI be less than 1 K to ensure that we are measuring a layer with an ash signal but, importantly, not an $SO_2$ signal. Similarly, we require that the Kasatochi and Sarychev layers only exhibit an $SO_2$ signal (SI $\geq$ 1 K) and do not exhibit an ash signal (AI $<$ 1 K). To make this point clear we have revised the relevant part of Section 3.1 as follows:

"For the Puyehue case study, this set of collocated AIRS pixels is scanned for an AI greater than or equal to 1 K and SI below 1 K. These conditions were set to ensure that the volcanic aerosol layers analysed for the Puyehue case study were dominated by an ash signal and, importantly, did not exhibit an $SO_2$ signal. Similarly, to ensure that observations of volcanic layers for the Kasatochi and Sarychev case studies were dominated by sulfates (and not an ash), the algorithm required an SI greater than or equal to 1 K and an AI below 1 K."

Section 3.2: What is meant by the 'mean scattering ratio'?

Response:
    Thank you for this comment. The authors meant to refer to the mean of the **attenuated** scattering ratio profile, $R'(r)$. The attenuated scattering ratio profile is defined as the ratio of the total attenuated backscatter profile, $\beta'(r)$, to the attenuated molecular backscatter profile, $\beta'_m(r)$ (Vaughan et al., 2009):

$$R'(r) = \frac{\beta'(r)}{\beta'_m(r)} = \frac{\beta'_m(r) + \beta'_p(r)}{\beta'_m(r)} \tag{3}$$

For the top layer in a given CALIOP profile, the two way transmittance constraint is calculated by taking the mean of $R'(r)$ in the clear air region immediately below the detected aerosol layer i.e. $T_e^2(r_t, r_b) = \langle R'_{below}(r) \rangle$, where the particulate backscatter is assumed to be zero. The clear air region is defined by the 'clear air analysis depth', which is determined via an iterative process in the SIBYL algorithm (see Sect. 4.3 of Vaughan et al., 2005). This description is included in Sect. 2.2 of the revised manuscript.

Section 4.2: Mean color ratio for Sarychev layer does not agree with value given in Table 2.

Response:
    This error has been corrected in the revised manuscript.

Section 4.2: In the Abstract it was reported that the depolarization ratio exponentially decreased with time. This is not reported in Section 4.2. As the change in the optical properties is an important point and thus reported in the Abstract it should also be referred to in the description of the Sarychev

layer. How do the other optical properties behave? What are the properties in the beginning of the observations (, mid) and end? This is reported quite lately in the manuscript. To which periods does the mean value correspond to? If the mean values are calculated from the whole period what is the significance of this mean value?

Response:

The authors agree that the decrease in the depolarization ratio with time for the Sarychev case study is an important finding. This finding has now been reported in Sect. 4.2. We have now reported on the change with time (beginning, middle, end) for each of the optical properties in Sects. 4.1, 4.2 and 4.3 in the revised manuscript. The mean values reported in Sect. 4 correspond to the whole time period for each case study. Given the positively skewed distributions shown in Fig. 2, we suggest that the median value is more significant than the mean values given for the case studies considered. This is now stated in the revised manuscript.

Section 4.3: How is a valid lidar ratio profile defined (time/length)? The number of cases/profiles resulting into the mean values should also be given for the other cases. Information about the CALIOP measurements (number, time, days, and location) should be given for the different cases should.

Response:

The lidar ratio retrievals are of a single value for any profile of attenuated backscatter and are constrained by the measurement of effective transmittance. It is not possible to retrieve a "lidar ratio profile" with that single constraint. Valid lidar ratio retrievals are those which satisfy constrained conditions i.e. that are constrained by an estimate of the effective two-way transmittance. We now explicitly define what we mean by 'valid' lidar ratio retrievals in Section 2.2:

"To ensure constrained conditions for the lidar ratio retrieval (i.e. clear air above and below a lofted layer with acceptable SNR), only stratospheric volcanic aerosol layers that had an extinction quality control flag equal to 1, a valid two-way transmittance measurement (i.e. $0 < T_e^2 < 1$) and a horizontal averaging value of 5 km were included in the analysis. We refer to 'valid' lidar ratio retrievals hereafter as having satisfied these criteria."

The number of (valid) lidar ratio retrievals resulting into the means are reported for each case study in the revised manuscript (Sections 4.1, 4.2 and 4.3). The time period for each case study and geographic region/locations analysed are also stated in these sections. The number of layers contributing to the mean geometric and optical properties are reported in Tables 1 and 2 of the revised manuscript and the specific measurement time periods for each case study are discussed Section 6.2 of the revised manuscript.

Section 4.3: Standard deviation of color ratio does not agree with the value given in Table 2.

Response:

This error has been corrected in the revised manuscript.

Section 5.1: What is meant by the aerosol scattering ratio?

Response:

The terms "aerosol scattering ratio" , "particulate scattering ratio", "backscatter ratio" and "scattering ratio" all appear in the literature and usually have the same definition. Here, the aerosol scattering

ratio, $R_p(r)$, reported in Vernier et al. (2009), is defined in the present notation as

$$R_p(r) = \frac{\beta_m(r) + \beta_p(r)}{\beta_m(r)}. \tag{4}$$

This is distinct from the attenuated scattering ratio, $R'(r)$, which has not been corrected for molecular, particulate and ozone attenuation. We refer to it as the particulate (aerosol) scattering ratio in the revised manuscript for consistency:

"Vernier et al. (2009) highlighted how this issue would impact the CALIOP calibration region, concluding that undetected aerosols up to 35 km lead to an underestimation of the particulate (aerosol) scattering ratio (an average relative error of 6%), with the effects most pronounced in the tropics (20°N–20°S)."

Section 5.3: Why did you use the error calculation according to Equation 9? This formula is used for the calculation of random (statistical) errors. To my understanding, the errors considered in this manuscript are not random errors and thus the total error should be calculated from the absolute error values of the considered parameters.

Response:
    Equation 9 is used in the standard procedure for calculating perturbation errors (see, for example, Chapter 4 of Hughes and Hase, 2010). We consider the errors discussed in Sect. 5 as being systematic i.e. they are errors that are constant through a given profile and cannot be reduced from averaging. This is the same definition used (and explained in detail) in Young et al. (2013). Specifically, we investigate how the errors in different key variables propagate into the lidar ratio ratio retrieval when they are perturbed. If it is assumed that the error in each perturbation variable is uncorrelated then the total error is calculated from the absolute errors by summing them together in quadrature (i.e. the square root of the sum of the squares of the errors). This is because we assume that the total error makes up an error surface composed of the independent component errors. Thus we use Pythagorus' theorem in $N$ dimensions to construct the total error from the component errors (Hughes and Hase, 2010).

Figures 5-7: Please indicate the aerosol free regions below and above the volcanic layer.

Response:
    Thank you for this comment. The regions above the layers are assumed to be aerosol free. We account for and discuss errors that may be introduced by this assumption in Sect. 5.2. We have now indicated the clear air regions below each layer on Figs. 5–7 of the revised manuscript.

Section 6.1: The mean values of the lidar ratio for the Kasatochi and Sarychev layers shown in this case studies are smaller than the mean values reported for the whole measurements for these layers. Can you give more information about the changes over the time? Maybe give a time series of the lidar ratio for the different volcanic layers to illustrate the changes and / or variability over time. Otherwise the mean values of the case studies or the mean values over all suffer the loss of significance.

Response:
    The purpose of Sect. 6.1 is to give the reader an idea for the spatial variation of lidar ratio across well-defined volcanic ash and sulfate layers. It also illustrates (Figs. 5–7 of original manuscript) the conditions under which the lidar ratio retrievals are successful and how the volcanic layers correlate

with the AI and SI. In the revised manuscript, we have added text to emphasise this point and have also discussed the time of the observations relative to the start of each eruption. We have also annotated where each of the selected observations correspond to on the time series plot given in the revised Fig. 8 (Figure 1 of this document). This figure also shows the time series of the color and depolarization ratios for each the three case studies.

Section 6.2: It is right that the measurements of the different volcanic layers correspond to different stages / ages of the volcanic layer. However the way it is described here could lead to misinterpretation of this information, as one could think that the measurements of the three volcanic layers can be related to each other and show an alteration of volcanic aerosol layers during time.

Response:
    The authors did not intend to give this impression. Section 6.2 has been revised to make it clear that the aerosol layers should not be related to each other directly in terms of aerosol evolution. The revised description is:

"As volcanic aerosol layers evolve and disperse into the atmosphere their microphysical properties are expected to change with time. The Kasatochi and Puyehue layers were observable for a duration of ∼12 days, while the Sarychev observations covered a time period of ∼17 days. Figures 8a–c show that all observations were made more than three days after eruption onset. The Kasatochi and Puyehue volcanic aerosols were observed for a similar time period (∼12 days); however, for the Puyehue case study, the aerosol layers had resided in the stratosphere for more than 11 days before the measurement period began. The Sarychev case study covered the longest observational time period, providing observations of sulfate-rich aerosols for over two weeks. All volcanic aerosol layers were subject to long-range transport across the globe as shown by the spatial distribution of observations plotted in Figs. 8j–l."

Section 6.2: The increase of the Puyehue ash layer with time is small compared to the uncertainties of the retrieved property, thus the statement derived from this changes is very speculative.

Response:
    The authors agree. Indeed, the revised version of Fig. 8 (Figure 1 of this document) shows that this statement is even more speculative than first thought. We have therefore removed it from the revised manuscript. The revised statement is

"The particulate lidar ratios for all three case studies were quite variable with time (Figs. 8a–c). Over these time scales (1–2 weeks) it is likely that the volcanic aerosol layers are mixing with ambient aerosol, resulting in fluctuations in the lidar ratio with time. Changes in the lidar ratio may also be a result of sampling different parts of an inhomogeneous aerosol cloud."

Section 6.3, discussion about high lidar ratios for Puyehue: Should not the loss of the large particles also be reflected in the depolarization ratio? No changes are obvious there.

Response:
    For the Puyehue case study, the ash layers had already resided in the atmosphere for ∼11 days before the CALIOP measurements were available. This means that the larger particles would have already sedimented out before the measurement period began (see Rose and Durant, 2009, for discussion on atmospheric residence times of volcanic ash). We therefore do not capture the fall out of larger particles in the depolarization ratio, but instead observe layers composed of small, irregular

(depolarising) ash particles.

Page 8, lines 7-8: This statement about the volume and the particle depolarization ratio is misleading.

Response:
    We assume the reviewer is referring to the statement on page 18 lines 7–8:

"Note that $\delta_v$ is not strictly a particle property, but for layers dominated by aerosols it can be used as a first approximation to the particulate depolarization ratio, $\delta_p$ (Wiegner et al., 2012)."

We agree and have removed it from the revised manuscript.

**References**

Fernald, F. G., Herman, B. M., and Reagan, J. A.: Determination of Aerosol Height Distributions by Lidar., J. Appl. Meteorol., 11, 482–489, 1972.

Hughes, I. G. and Hase, T. P. A.: Measurements and their Uncertainties: A Practical Guide to Modern Error Analysis, Oxford University Press, 2010.

Platt, C.: Lidar and Radiometric Observations of Cirrus Clouds, J. Atmos. Sci., 30, 1191–1204, 1973.

Rose, W. I. and Durant, A. J.: Fine ash content of explosive eruptions, J. Volcanol. Geoth. Res., 186, 32–39, 2009.

Vaughan, M. A., Winker, D. M., and Powell, K. A.: CALIOP Algorithm Theoretical Basis Document Part 2: Feature Detection and Layer Properties Algorithms [Available online at `http://www-calipso.larc.nasa.gov/resources/pdfs/PC-SCI-202_Part2_rev1x01.pdf`], 2005.

Vaughan, M. A., Powell, K. A., Kuehn, R. E., Young, S. A., Winker, D. M., Hostetler, C. A., Hunt, W. H., Liu, Z., McGill, M. J., and Getzewich, B. J.: Fully Automated Detection of Cloud and Aerosol Layers in the CALIPSO Lidar Measurements, J. Atmos. Ocean. Tech., 26, 2034–2050, 2009.

Vernier, J. P., Pommereau, J. P., Garnier, A., Pelon, J., Larsen, N., Nielsen, J., Christensen, T., Cairo, F., Thomason, L. W., Leblanc, T., and McDermid, I. S.: Tropical stratospheric aerosol layer from CALIPSO lidar observations, 114, D00H10, 2009.

Wiegner, M., Gasteiger, J., Groß, S., Schnell, F., Freudenthaler, V., and Forkel, R.: Characterization of the Eyjafjallajökull ash-plume: Potential of lidar remote sensing, Phys. Chem. Earth, 45-46, 79–86, 2012.

Young, S. A., Vaughan, M. A., Kuehn, R. E., and Winker, D. M.: The Retrieval of Profiles of Particulate Extinction from Cloud–Aerosol Lidar and Infrared Pathfinder Satellite Observations (CALIPSO) Data: Uncertainty and Error Sensitivity Analyses, J. Atmos. Ocean. Tech., 30, 395–428, 2013.

---

## Author Comment (AC4) · 17 May 2017

**Response to RC4**

In the following response, reviewer comments are in black and author responses are in blue.

**General**

The paper is well written. Carefully analysed CALIOP observations are presented. The paper is appropriate for ACP.

The only negative and confusing point is that obviously the volume depolarization ratio and volume color ratio are used instead of the particle depolarization ratio and particle color ratio. But I am not sure what is shown. The authors have to clarify that when discussing equations 1 and 2, see details.

Minor revisions are at least required. However, major revisions (switch to particle depolarization ratio) would significantly improve the paper.

**Response:**

The authors thank the reviewer for their comments on the manuscript. As suggested, we have now included the particulate depolarization ratios. The 1064 nm lidar ratio ( $S_{p,1064}$ ) and layer-effective particulate color ratio ( $\chi_p$ ) can be simultaneously retrieved using the two-color method of Vaughan (2004). We went to considerable effort to set up such an analysis scheme to perform the calculations. However, we found that the method was rather insensitive to variations in  $S_{p,1064}$  because of the relatively weak signals and low optical depths of the volcanic aerosol layers. We therefore decided that these results added nothing to the value of the paper. We have added this comment in the revised manuscript as follows:

"We also note that the layer-effective particulate color ratio,  $\chi_p$ , can be retrieved using the two-color method of Vaughan (2004). This approach seeks to minimise a non-linear function by simultaneously varying  $S_{p,1064}$  and  $\chi_p$  using the method of non-linear least squares. However, for the case studies considered here, we found that the method was rather insensitive to variations in the 1064 nm particulate lidar ratio; often resulting in non-physical solutions for  $S_{p,1064}$ . We expect that this was due to the relatively weak signals and low optical depths of the volcanic aerosol layers under examination. As these results were inconclusive, and require a more complete treatment of the sources of error, we decided this analysis was outside of the scope of the present analysis and therefore do not report the results here."

Upon implementing the  $S_{p,1064}$  retrieval code we noticed an error in the  $S_{p,532}$  retrieval. The error was due to the way the initial lidar ratio (defined by Eq. (7) in the original manuscript) was calculated. In the original code,  $\eta$  values of 0.6 were used in Eq. (7) and  $\eta$  values of 0.90 (for Puyehue) and 0.95 (for Kasatochi and Sarychev) were used in Eq. (3) when we should have been using the same  $\eta$  values in both Eq. (7) and Eq. (3). We have now corrected this error by using an  $\eta$  value of 0.95 for Kasatochi and Sarychev and 0.90 for Puyehue in both Eqs. (3) and (7). We have found that this error resulted in lidar ratios that were biased high by ~4%. To illustrate this, we have plotted the original dataset against the  $\eta$  corrected dataset in Figure 1 of this document.

During this process we also found a bug in the lidar ratio retrieval code. The bug was due to the way the trapezoidal integration procedure (used to evaluate the integral term in the denominator of Eq. (3)) handled masked values. Specifically, if there was at least one masked value in an array then the integral of the array would be evaluated as being masked; leading to a masked lidar ratio retrieval, which was rejected from the analysis. We have revised the code now so that an

Figure 1: Comparison of  $S_p$  for the original dataset and the dataset corrected for the  $\eta$  error.

array containing masked values will still be evaluated. This is achieved using the cumulative trapezoidal integration module from the Scipy library (https://docs.scipy.org/doc/scipy/reference/ generated/scipy.integrate.cumtrapz.html). The result of this revision on the analysis is that more data points (more valid lidar ratio retrievals) are now analysed. The results presented in the revised manuscript do not, however, differ significantly from the results presented in the original manuscript and so the main conclusions drawn from the original manuscript have been retained. The impact of this correction on the analysis is shown for a specific example of an observation of an ash layer for the Puyehue case study (Figure 2 of this document). Here the added data points are in red and lidar ratios that have been corrected for the  $\eta$  error are in blue (Figure 2d of this document). Figure 3 (of this document) shows how the correction impacts the overall lidar ratio PDFs. In the revised manuscript, Figs. 2–10 of the original manuscript have been corrected for the  $\eta$  error and the integration bug (corresponding to  $\eta$  + integration values that are annotated on the subplots of Figure 3 of this document). The values in Tables 1–3 have also been corrected in the revised manuscript.

Details:

Abstract:

P1, L9: Please state the wavelength (532 nm) again in the case of the volume depolarization ratio.

**Response:**

Accepted.

P1, L10-12: A volume depolarization ratio of 0.08, 0.05, 0.25 tells us almost nothing as long as we do not know the backscatter ratio (total-to-Rayleigh backscatter). So again, why not trying to determine the particle depolarization ratio? At least for a few examples.

**Response:**

The authors disagree that the volume depolarization ratios tell us "almost nothing" without the scattering ratio. The volume depolarization ratios presented do show distinctions between the layers identified as sulfates and the layers identified as volcanic ash (Fig. 9 of the original manuscript). One could argue that, for CALIOP, the volume depolarization ratios are more useful than the particulate depolarization ratio as the volume depolarization ratios are direct measurements (i.e. do not require